# Reinforcement Learning of Adaptive Acquisition Policies for Inverse Problems

## Abstract

A promising way to mitigate the expensive process of obtaining a high-dimensional signal is to acquire a limited number of low-dimensional measurements and solve an under-determined inverse problem by utilizing the structural prior about the signal. In this paper, we focus on adaptive acquisition schemes to save further the number of measurements. To this end, we propose a reinforcement learning-based approach that sequentially collects measurements to better recover the underlying signal by acquiring fewer measurements. Our approach applies to general inverse problems with continuous action spaces and jointly learns the recovery algorithm. Using insights obtained from theoretical analysis, we also provide a probabilistic design for our methods using variational formulation. We evaluate our approach on multiple datasets and with two measurement spaces (Gaussian, Radon). Our results confirm the benefits of adaptive strategies in low-acquisition horizon settings.

## 1 Introduction

Compressed sensing aims at solving underdetermined linear inverse problems by leveraging the structure of the underlying signal of interest (Tibshirani, 1996; Candès et al., 2006; Donoho, 2006). Although the initial theory was focused on sparsity, other notions of structure have been considered too, for instance, in Tang et al. (2013). Compressed sensing theory provides a hard constraint on the number of required measurements for signal recovery. Reducing the number of measurements is crucial when they are costly, for example, due to resource constraints or patient comfort in imaging tasks, and there has been a line of works exploring adaptive measurements to achieve this goal (see related works section). These works have been mainly focused on medical imaging applications like MRI where the space of measurements is a discrete space (see for example Bakker et al. (2020) and references therein).

On the other hand, certain works in compressed sensing theory suggest adaptive methods are not helpful in noiseless settings when the worst-case error is considered (Cohen et al., 2009; Foucart et al., 2010; Novak, 1995). At first look, this seems to be in conflict with the experimental gains shown by the adaptive methods. It is important to understand the roots of this discrepancy and see if any guidelines can be obtained by revising the existing theoretical results.

In this work, we pursue two goals. First, we aim to design a generic framework to solve adaptive recovery problems by simultaneously learning a measurement policy network and a recovery algorithm , while working with either continuous or discrete measurement spaces. Moreover, we aim to explain the apparent discrepancy between certain theoretical results and the experimental works and derive design guidelines. We stress that our method only needs to learn the measurement *actions* and the recovery network. Thereby, our model can potentially be agnostic to the exact measurement model, making it a suitable candidate for non-linear settings.

**Contributions.** **1)** We introduce a framework for adaptive sensing with arbitrary sensing operations that can naturally work in both continuous and discrete spaces. **2)** We propose a novel training procedure for end-to-end learning of both reconstruction and acquisition strategies, which combines supervised learning of the signal to recover and reinforcement learning on latent space for optimal measurement selection. **3)** We add a probabilistic formulation of our model, which can be trained with a variational lower bound to add structure and probabilistic interpretation to the latent space.

## 2 Methodology

### 2.1 Compressed Sensing

**Problem.** As in compressed sensing, we consider underdetermined inverse problems, where the goal is to recover a high-dimensional *signal* $\boldsymbol{x} \in \mathbb{R}^N$ through low-dimensional *observations* $\boldsymbol{y} \in \mathbb{R}^T (T \ll N)$. Each observation $y_t = F(\boldsymbol{a}_t, \boldsymbol{x})$ is acquired through a projection operation parameterized as $\boldsymbol{a}_t$. In a linear measurement setting, this amounts to:

$$\boldsymbol{y} = F(\boldsymbol{A}, \boldsymbol{x}) = \boldsymbol{A}\boldsymbol{x} \tag{1}$$

where $\boldsymbol{A} \in R^{T \times N}$ is referred to as the 'sensing matrix', and is defined as:

$$\boldsymbol{A} = \begin{bmatrix} \boldsymbol{a}_1 \\ \boldsymbol{a}_2 \\ \vdots \\ \boldsymbol{a}_T \end{bmatrix} \tag{2}$$

We will explain specific types of $\boldsymbol{A}$, $\boldsymbol{x}$, $y$, and $F$ used in this work in section 4.1.

**Goal 1: Reconstruction.** The primary goal in the compressed sensing setup is to recover the signal $\boldsymbol{x}$. Note that this reconstruction is generally intractable as it amounts to solving an under-determined system of equations. However, the reconstruction becomes possible if we assume a prior about the signal structure, for example, the assumption of sparsity or lying on the data manifold modeled by a deep generative model (Bora et al., 2017).

**Goal 2: Reducing Measurements.** An auxiliary goal in compressed sensing is to reduce the number of measurements (i.e., the number of rows in $\boldsymbol{A}$) with minimal impact on the recovery process. This is especially critical when the measurement process is an expensive, time-consuming process, such as with medical MRI or CT scans. Reducing the measurements can be achieved by designing a better measurement matrix $\boldsymbol{A}$.

The above two goals present an inherent trade-off: we can obtain better reconstructions at the price of collecting more expensive, time-consuming measurements. In the next section, we present our technique that jointly addresses this trade-off.

### 2.2 Adaptive Compressed Sensing

**Framework.** Central to our approach towards minimizing the number of measurements is exploiting adaptivity: to sequentially construct the sensing matrix $\boldsymbol{A}$ (composed of $T$ projection vectors $\boldsymbol{a}_t$) to enable better recovery of the underlying signal $\boldsymbol{x}$. Similar to Bakker et al. (2020), we approach this sequential decision-making problem within a reinforcement learning framework. In the following, we use the terms active and adaptive strategies interchangeably, meaning strategies that select custom measurements depending on the specific input or signal. Other works, such as Bakker et al. (2022), make a distinction between active and adaptive, with adaptive meaning a custom set of measurements per input collected all at once, and active meaning that several rounds of measurements and observations are done, potentially also more than one measurement at the time like in Yin et al. (2021).

**Adaptive Acquisition as a POMDP.** We define the adaptive acquisition as a Partially Observable Markov Decision Process (POMDP). A POMDP is defined by a tuple $(\mathcal{S}, \mathcal{A}, \mathcal{O}, \mathcal{F}, \mathcal{U}, \mathcal{R})$. Here, $\mathcal{S}$ is the state space, $\mathcal{A}$ the action space, $\mathcal{O}$ the observation space, $\mathcal{F} : (\mathcal{S} \times \mathcal{A}) \to \mathcal{S}$ is the transition distribution, $\mathcal{U} : (\mathcal{S} \times \mathcal{A}) \to \mathcal{O}$ is the observation distribution, and $\mathcal{R} : (\mathcal{S} \times \mathcal{A}) \to \mathbb{R}$ is the reward function for a state-action pair. Over the next paragraphs, we take a closer look at each of these aspects of the problem of adaptive compressed sensing.

**Stationary State and Transition Distribution.** We consider the signal $\boldsymbol{x}$, which needs to be recovered as the stationary state of the system. The agent cannot directly observe the state but rather senses and obtains low-dimensional observations $y_t$. As a result of the stationary state, the transition distribution remains fixed:

$$\mathcal{F}(\boldsymbol{x}_{t+1} \mid \boldsymbol{x}_t, \boldsymbol{a}_t) = \delta(\boldsymbol{x}_{t+1} - \boldsymbol{x}_t), \quad \boldsymbol{x}_0 = \boldsymbol{x} \tag{3}$$

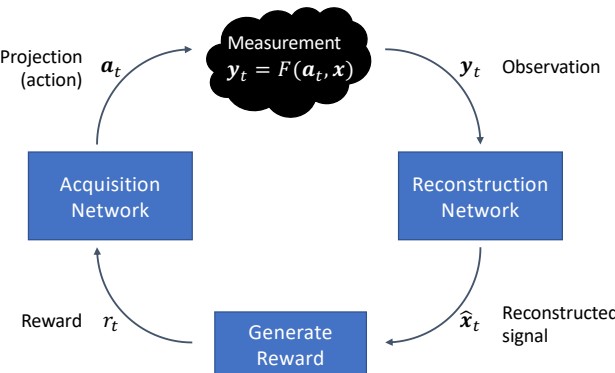

Figure 1: Schematic representation of our method. A reconstruction network is trained to reconstruct the signal $x$ given a sequence of actions $a_{1:t}$ and corresponding observations $y_{1:t}$. The role of the acquisition network is to select the next action $a_{t+1}$ based on the reconstruction quality of the signal $\hat{x}_t$. The improvement in reconstruction quality between consecutive steps $t$ and $t-1$ is used as reward $r_t$ to train the acquisition network with Reinforcement Learning. After a new action $a_{t+1}$ is selected, a new observation $y_{t+1}$ is collected based on a function $F(a_{t+1}, x)$, specific to the inverse problem at hand. Note that in real-world scenarios, there might be no knowledge of $F$ and $x$, and the observation $y$ can be obtained only through measurements $a$ of the environment.

which would mean that $\boldsymbol{x}_{t+1} \sim \delta(\boldsymbol{x}_{t+1} - \boldsymbol{x})$, where $\delta$ is the Dirac distribution.

**Actions.** Each action $\boldsymbol{a}_t$ corresponds to a particular projection operation, i.e., a row of the sensing matrix $\boldsymbol{A}$. Depending on the type of measurement used (more details in 4.1), the actions can take different forms but generally are $N$-dim real-valued vectors.

**Observations and Observation Distribution.** We denote as *observation* $y_t$ the information received by the agent after performing a measurement $y_t = F(\boldsymbol{a}_t, \boldsymbol{x})$. This observation is drawn from observation distribution $y_t \sim \mathcal{U}(y_t | \boldsymbol{x}, \boldsymbol{a}_t)$. Since the paper primarily studies reconstructing a time-invariant signal in a noiseless setting, the observation corresponds to a deterministic measurement $y_t = F(\boldsymbol{a}_t, \boldsymbol{x})$.

**Reward.** After taking an action, the agent receives a reward according to the distribution $r_t \sim \mathcal{R}(r_t \mid \boldsymbol{x}, \boldsymbol{a}_t)$. We define the reward at each time step as the improvement in reconstruction quality $d(.)$ between consecutive time steps: $r_t = d(\hat{\boldsymbol{x}}_t, \boldsymbol{x}) - d(\hat{\boldsymbol{x}}_{t-1}, \boldsymbol{x})$ following a metric $d$. In our experiments, we use Structural Similarity Index Measure (SSIM) (Wang et al., 2004) as $d(.)$. For $t=1$, the reward is simply $d(\hat{\boldsymbol{x}}_1, \boldsymbol{x})$.

### 2.3 Architectural components

Central to our approach (see Figure 1) are two models: (a) a *reconstruction* model that recovers the signal from low-dimension observations; and (b) an *acquisition* model (the agent) that adaptively constructs the sensing matrix for measurements. We now look into these models in-depth.

**Reconstruction Model.** The goal of the reconstruction model (see Figure 2; in blue) is to recover a signal $\hat{\boldsymbol{x}}_t$ that faithfully represents the signal $\boldsymbol{x}$ under measurement. Such a recovery is challenging, given that the system is under-determined, i.e., we recover using $T$ measurements $\boldsymbol{y}_{1:T}$ while the signal is $N$-dimensional ($N \gg T$). To tackle this challenge, inspired by Bakker et al. (2020), we use an autoencoder-style model to perform reconstruction but with one key difference: the encoder is a Gated Recurrent Unit (GRU) (Cho et al., 2014) designed to deal with variably-sized sequences. The autoencoder is defined by a (recurrent) encoder $g_\phi$ and a decoder $f_\phi$, both parameterized by $\phi$ (Fig. 2). At each time-step $t$, the encoder predicts the latent features $z_t$ from the trajectory of actions $\boldsymbol{a}_{1:t}$ and observations $\boldsymbol{y}_{1:t}$: $\boldsymbol{z}_t = g_\phi(\boldsymbol{a}_t, \boldsymbol{y}_t, \boldsymbol{h}_t)$, where $\boldsymbol{h}_t$ is the hidden state of the GRU and summarizes the past inputs $a_{1:t-1}$ and $y_{1:t-1}$. The decoder receives $\boldsymbol{z}_t$ as input and outputs a reconstruction $\hat{\boldsymbol{x}}_t$: $\hat{\boldsymbol{x}}_t = f_\phi(\boldsymbol{z}_t)$.

The model is trained to minimize a loss $\mathcal{L}$ defined as the sum of the Mean Squared Error (MSE) between $x$ and $\hat{x}_t$ for each $t$:

$$\mathcal{L} = \frac{1}{T} \sum_{t=1}^{T} (\boldsymbol{x} - \hat{\boldsymbol{x}}_t)^2 \qquad (4)$$

As opposed to solely considering the MSE with the final reconstruction $\hat{\boldsymbol{x}}_T$, our formulation presents certain benefits: (a) it takes into account the scenario in which $\boldsymbol{x}$ changes over time (e.g., when taking a measurement affects the true signal); (b) it forces the model to have good reconstruction quality at each time step; and (c) makes the loss comparable to the variational evidence lower bound optimized in section 4.4, which includes the sum of likelihoods at each time step of the trajectory.

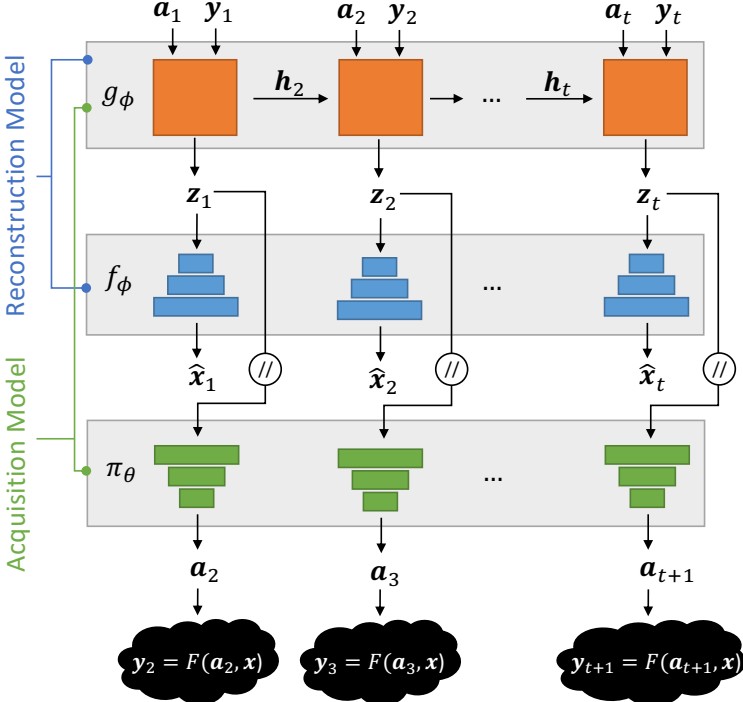

Figure 2: Network architecture used in our experiments. A recurrent encoder (orange) maps the action $\boldsymbol{a}_t$ and observation $\boldsymbol{y}_t$ at time step $t$ to a latent representation $\boldsymbol{z}_t$, using the hidden state $\boldsymbol{h}_t$ to summarize the past actions and observations $\boldsymbol{a}_{1:t-1}$ and $\boldsymbol{y}_{1:t-1}$. A convolutional decoder is then used to reconstruct the signal $\hat{\boldsymbol{x}}_t$ from $\boldsymbol{z}_t$. The acquisition network is used to select actions $\boldsymbol{a}_{t+1}$ from the latent representation $\boldsymbol{z}_t$. The acquisition network is only used for the adaptive acquisition strategies, while the random baseline samples actions at random from a predefined probability distribution. Note how the encoder only receives gradients from the decoder $f_\phi$, and gradients from the acquisition network are never backpropagated through the encoder.

**Acquisition Model (Policy Network).** The goal of the acquisition model is to predict a projection (the action) $\boldsymbol{a}_t$, such that the resulting observation $\boldsymbol{y}_t = F(\boldsymbol{a}_t, \boldsymbol{x})$ is highly informative towards reconstructing the signal $\hat{\boldsymbol{x}}_t$. To achieve this goal, we design a policy $\pi_\theta$ (see Figure 2; in green) to select the next action based on the history of measurements, observations, and reconstruction qualities. Therefore, we condition the policy network on the encoder's latent representation $\boldsymbol{z}_{t-1} = g_\phi(\boldsymbol{a}_{t-1}, \boldsymbol{y}_{t-1}, \boldsymbol{h}_{t-1})$, which is in parallel also used to aid reconstruction as we saw earlier. Following the manifold hypothesis (Pope et al., 2021; Fefferman et al., 2016; Bengio et al., 2013), we assume that $\boldsymbol{z}_{t-1}$ contains all the relevant information about the history of acquisitions and observations in a compressed space. Therefore, the policy learns to select the next action

$\boldsymbol{a}_{t+1}$ conditioned only on the latest latent representation:

$$\boldsymbol{a}_t \sim \pi_\theta(\boldsymbol{a}_t \mid \boldsymbol{z}_{t-1}) \tag{5}$$

This differs from what is commonly done in Adaptive Sensing for MRI, where the policy is conditioned on the latest reconstruction $\hat{x}_t$ (Bakker et al., 2020). Following this acquisition, we reconstruct the signal based on the latest observation to assign an instantaneous reward $r_t$. We train the acquisition model with Vanilla Policy Gradient (VPG) (Sutton and Barto, 2018), specifically using *reward-to-go* $\hat{R}_t = \sum_{k=t}^{T} \gamma^{k-t} r_k$ with discount factor $\gamma$, advantage estimation $A^\pi$ (Schulman et al., 2016), and a neural network $V_\psi(z_t)$ as baseline for variance reduction. We estimate the policy gradient for a batch of $B$ images as:

$$\hat{g}_B = \frac{1}{B} \sum_B \sum_{t=1}^{T} \nabla_\theta \log \pi_\theta(\boldsymbol{a}_t \mid \boldsymbol{s}_t) \hat{A}_t \tag{6}$$

with $\theta$ the parameters of the policy network. The baseline (subsection 4.2) is trained by mean squared error as:

$$\psi_{k+1} = \arg\min_\psi \frac{1}{BT} \sum_B \sum_{t=1}^{T} \left( V_\psi(\boldsymbol{z}_t) - \hat{R}_t \right)^2 \tag{7}$$

## 2.4 Training strategy and Baselines

In this section, we walk through our end-to-end approach where the reconstruction and adaptive acquisition model are jointly trained. We begin discussing two baseline approaches and conclude with the end-to-end approach.

**Baseline 1: Random Acquisition and Reconstruction (`AE-R`).** In this baseline, we consider a random acquisition policy: each action taken is drawn agnostic to past actions, observations, and rewards. Note that in this case, there is no inherent concept of temporal sequences, as all the measurements are done in parallel and independently of each other, therefore removing the need for a recurrent encoder. However, to make the baseline comparable with other acquisition strategies with sequential acquisition, we keep the same architectural components described in the previous section.

**Baseline 2: Adaptive Acquisition with pre-trained Reconstruction (`AE-P`).** The second baseline consists of a pre-trained reconstruction model, trained with the same procedure used for the `AE-R` baseline. This strategy is similar to that proposed in Bakker et al. (2020). The policy selects a first measurement $\boldsymbol{a}_1$, which is used to obtain a first observation $y_1$. Then, for $t = 1, \ldots, T$, we compute $\boldsymbol{z}_t = g_\phi(\boldsymbol{a}_t, \boldsymbol{y}_t, \boldsymbol{h}_t)$ ($\boldsymbol{h}_1$ is filled with zeros) and $\hat{\boldsymbol{x}}_t = f_\phi(\boldsymbol{z}_t)$. The policy selects the next action $\boldsymbol{a}_{t+1}$ based on $\boldsymbol{z}_t$, and the procedure is repeated until the entire trajectory is collected. Finally, the policy is updated with Policy Gradient (see section 2.3), while the reconstruction model is kept fixed. The policy network is always trained on the same data used to pre-train the reconstruction network.

**End-to-End Adaptive Acquisition and Reconstruction (`AE-E2E`).** We proposed end-to-end training of both reconstruction and acquisition models. Unlike the previous `AE-P` strategy, the reconstruction model is now initialized with random parameters and is trained after each trajectory is collected by the policy. Inspired by Zintgraf et al. (2020), we do not backpropagate gradients from the policy to the encoder, to avoid instabilities and the need for additional hyperparameters for the multi-task learning loss.

## 2.5 Bayesian reasoning and the importance of belief states

In this section, we extend our approach with a variational formulation (Kingma and Welling, 2014) to reap two additional benefits: (a) generalizability, as simple auto-encoders learning an unstructured latent space, typically leads to overfitting (Chung et al., 2015); and (b) uncertainty quantification, which provides a reasonable signal to guide the policy network. Intuitively, we expect the policy to select exploratory actions in high uncertainty regimes and more exploitative actions as uncertainty reduces. Consequently, we introduce a variational formulation for our model, inspired by Zintgraf et al. (2020), to learn a belief distribution over

the latent space. The encoder outputs the parameters of such distribution, in our experiments, a Multivariate Gaussian with diagonal covariance like in standard Variational Autoencoders (Kingma and Welling, 2014):

$$\boldsymbol{b}_t = g_\phi(\boldsymbol{a}_t, \boldsymbol{y}_t, \boldsymbol{h}_t) = (\bar{\boldsymbol{\mu}}_t, \bar{\boldsymbol{\sigma}}_t) \tag{8}$$

The policy selects the following action based on the belief:

$$\boldsymbol{a}_{t+1} = \pi_\theta(\boldsymbol{a}_{t+1} \mid \boldsymbol{b}_t), \tag{9}$$

while the decoder reconstructs the image from a sample from the belief distribution:

$$\hat{\boldsymbol{x}}_t = f_\phi(\boldsymbol{z}_t), \ \boldsymbol{z}_t \sim \mathcal{N}(\bar{\boldsymbol{\mu}}_t, \bar{\boldsymbol{\sigma}}_t I) \tag{10}$$

We train the model to maximize the probability of the signal $\boldsymbol{x}$ given the sequence of acquisitions and observations:

$$\log p(\boldsymbol{x} \mid \boldsymbol{a}_{1:T}, \boldsymbol{y}_{1:T}) = \log \int p(\boldsymbol{x}, \boldsymbol{z}_{1:T} \mid \boldsymbol{a}_{1:T}, \boldsymbol{y}_{1:T}) d\boldsymbol{z}_{1:T} \tag{11}$$

As this probability is intractable, we maximize the ELBO instead:

$$\sum_{t=1}^{T} \mathbb{E}_{z_{1:T-1}} \left[ \mathbb{E}_{z_T} [\log p(\boldsymbol{x}_t = \boldsymbol{x} \mid \boldsymbol{z}_t)] - D_{KL}(q(\boldsymbol{z}_t \mid \boldsymbol{a}_t, \boldsymbol{y}_t, \boldsymbol{h}_t) \parallel p(\boldsymbol{z}_t \mid \boldsymbol{z}_{t-1})) \right] \tag{12}$$

The lower bound derivation can be found in Appendix B. The training procedure remains the same as for the deterministic end-to-end case. The prior at $t = 1$ is $p(\boldsymbol{z}_1) = \mathcal{N}(0, I)$, while at each time step $t > 1$ it is the posterior $q$ at time $t - 1$.

We would like to end this section with a remark on the theoretical support for the gain of adaptive acquisition. In compressed sensing, there are certain results that state there is no gain in adaptive sensing (see for example Foucart et al. (2010); Foucart and Rauhut (2013)). We analyze the assumptions behind the theory in App. A. To summarize, first, the adaptive sensing does not improve the worst-case error, and second, the theory does not apply to a probabilistic adaptive scheme. In this work, we have focused on average-case error improvements, and used a probabilistic formulation of adaptive sensing. See App. A for detailed discussions.

## 3 Related Work

### 3.1 Adaptive acquisition with Reinforcement Learning

The question of adaptive compressed sensing has been approached from a theoretical perspective in Cohen et al. (2009); Foucart et al. (2010) (see Foucart and Rauhut (2013) for a concise and detailed overview of arguments based on Gelfand width). There are other works discussing various methods and theoretical analysis for the adaptive sensing (Malloy and Nowak, 2014; Castro, 2014; Castro and Tánczos, 2015; Davenport et al., 2016; Braun et al., 2015). These works consider in general noisy setting, while we are focused on noiseless setting here. Their approach is classical and not data-driven. In this work, we focus on Gelfand width-based analysis and review the subtleties of this argument in App. A and generalize some of the existing results. Complementing traditional compressed sensing approaches which focus on solving the *reconstruction* problem, we consider adaptive sensing approaches (Bakker et al., 2020; Pineda et al., 2020; Jin et al., 2019; Bakker et al., 2022; Ramanarayanan et al., 2023) where *acquiring* measurements are treated as a sequential decision-making problem. While our approach is closely related to the latter line of adaptive sensing techniques, we present a more general approach that is suitable beyond MRI scenarios, e.g., capable of dealing with both continuous and discrete sensing matrices and observations. More importantly, we tackle both reconstruction and acquisition problems simultaneously thereby contrasting the two-stage training procedure, in which a model is first trained for reconstruction and subsequently for acquisition. While the authors in Yin et al. (2021) also tackle these problems simultaneously, their approach trains the whole system in a supervised manner with short acquisition horizons (4 steps in their experiments). To the best of our

knowledge, we are the first to introduce an end-to-end training of reconstruction and acquisition models for active sensing with reinforcement learning. Our method also extends the work of Zintgraf et al. (2020), which proposes a method to learn a belief distribution over unknown environments. While we use a similar architecture and training procedure, our work differs in its goal. The work Zintgraf et al. (2020) performs meta-learning over the transition probability and reward function of unknown environments while we train our models to extract a belief distribution over the unknown state of a POMDP, similarly to Igl et al. (2018); Lee et al. (2020). In addition, the decoder in Zintgraf et al. (2020) is used to predict the future (and the past) of a Bayesian Augmented MDP, while for us, it is used as a reconstruction network, which is crucial to our end goal, namely to achieve a reconstructed signal faithful to the original signal $\boldsymbol{x}$.

### 3.2 Latent Variable models and Deep RL

The papers Chung et al. (2015); Gregor et al. (2018) introduce variational methods for modeling temporal sequences, arguing that it is important to have a latent representation that can form a belief distribution representing a measure of uncertainty. We draw inspiration from these works to introduce a probabilistic interpretation of the latent space in our model. Our architecture and training procedure are similar to the ones used in Reinforcement Learning on latent space methods (Khan et al., 2019; Allshire et al., 2021; Zhou et al., 2021), Meta Reinforcement Learning (Wang et al., 2016; Duan et al., 2016; Zintgraf et al., 2020) and Model-Based Reinforcement Learning (Hafner et al., 2020; 2021). In the context of POMDPs, works such as Igl et al. (2018); Hafner et al. (2019); Lee et al. (2020) use Deep Variational Reinforcement Learning to learn a belief distribution over the hidden state, showing how explicitly modeling the uncertainty improves the performance of the agent. In our work, such belief distribution must be suitable for both the acquisition and reconstruction networks to select the follow-up action and reconstruct the unknown signal $\boldsymbol{x}$.

## 4 Experiments

In this section, we describe the experiments performed to evaluate the proposed method. We first describe the datasets used, and then provide a comparison of the three methods introduced above, AE-R, AE-P, and AE-E2E. Finally, we verify the effectiveness of the variational formulation introduced in section 2.5. Additional experiments can be found in Appendix D.2, where we analyze a phenomenon observed in Bakker et al. (2020), where greedy policies ($\gamma = 0$) seem to perform better than discounted ones. Note that we focus on continuous action spaces. Moreover, we report additional ablation studies and complementary results for some of the datasets in Appendix D.4 and D.5. While our method can in principle deal with non-linear CS problems, we do not experiment with those in this work.

It is important to note that the use of adaptive acquisitions is particularly relevant when the acquisition budget is low. In principle, if one could take infinite measurements, then there wouldn't be any benefit in using an adaptive strategy over a random one. In the following, we provide an evaluation for different tasks at different measurement budgets, which can be used as a guideline to understand when the proposed adaptive strategy should be the preferred alternative over random acquisitions. In practice, the acquisition horizon $T$ is often driven by the specific domain, and the choice of acquisition strategy should be selected accordingly.

### 4.1 Datasets and Sensing Operations

We test our algorithms for adaptive acquisition on two datasets: the handwritten digits dataset MNIST (Deng, 2012) and the Low Dose CT Image and Projection Data[1] (MAYO) dataset (Moen et al., 2021) (more details in App. C.1). We use two different sensing operations $F$, which we name as *Gaussian* and *Radon*. The Gaussian transformation, denoted as $G$, consists in a matrix multiplication between a random Gaussian matrix $\boldsymbol{A}$ and the flattened image $\boldsymbol{x}$: $\boldsymbol{y} = G(\boldsymbol{A}, \boldsymbol{x}) = \boldsymbol{A}\boldsymbol{x}$. The rows $\boldsymbol{a}_t$ of the sensing matrix have continuous entries $\in (-\infty, \infty)$, and same for $\boldsymbol{y}$. For a vectorized image of size $N \times 1$, the sensing matrix has size $T \times N$ (each row $\boldsymbol{a}_t$ is $1 \times N$, with $T$ the total number of acquisitions, and $\boldsymbol{y}$ is a vector of dimensions $1 \times T$. In the adaptive acquisition scenario, the resulting $y_t = G(\boldsymbol{a}_t, \boldsymbol{x})$ is a scalar. Note that in compressed sensing, Gaussian measurements can achieve theoretical limits for non-adaptive sensing, and therefore represent the

---

[1] https://www.aapm.org/grandchallenge/lowdosect/

best non-adaptive sensing scheme. The Radon transform is commonly used to reconstruct images from CT scans, see Beatty (2012) for a detailed description. The sensing matrix $\boldsymbol{A}$ corresponds to a vector $1 \times T$ of angles in radians, with each $a_t$ being a scalar $\in [-\pi, \pi]$. Assuming that $x$ is an image of dimensions $h \times w$, then the $\boldsymbol{y}$ resulting from $R(\boldsymbol{A}, \boldsymbol{x})$ is a matrix of dimensions $h \times T$. In adaptive acquisition, $y_t = R(a_t, \boldsymbol{x})$ is a vector of dimensions $h \times 1$.

## 4.2 Implementation Details

In these experiments, we use a simple GRU encoder and convolutional decoder. The policy is a convolutional architecture like the decoder for the Gaussian measurements, while it is a Multi-Layer Perceptron (MLP) for Radon measurements. The value network baseline is always an MLP with one hidden layer and ReLU activation function. The MLP takes as input the latent vector from the encoder and outputs a scalar representing the baseline value used for variance reduction. All the policy models are trained with VPG and $\gamma = 0.9$ unless otherwise specified. For AE-R, the actions are randomly sampled from a spherical Gaussian $\boldsymbol{a}_t \sim \mathcal{N}(0, I)$ for Gaussian measurements and from a uniform $a_t \sim \mathcal{U}(-\pi, \pi)$ for Radon. For the adaptive acquisition, for both AE-P and AE-E2E models, the policy parametrizes the mean and standard deviation of a Gaussian distribution for Gaussian measurements, and the mean and concentration of a Von Mises distribution for Radon measurements. During training, actions are sampled from such distributions, while at validation the mean parameter is used as an action. More details about hyperparameters can be found in appendix C.

## 4.3 Random vs Adaptive Acquisitions

We start by comparing the performance of AE-R, AE-P, and AE-E2E models introduced in section 2.4 on the MNIST dataset, for both Gaussian and Radon sensing operations. We experiment with different trajectory length, $T = 20, 50, 100$ for Gaussian and $T = 5, 10, 20$ for Radon. This choice is motivated by Radon measurements providing more information than Gaussians (a vector instead of a scalar). The results are reported in table 1. We also report the reconstruction quality in SSIM after each acquisition step for models trained to optimize the whole trajectories, in Figure 4.

Table 1: Results on the MNIST datset for both Gaussian and Radon measurements in SSIM (higer is better). The trajectory length of the experiment is reported on the second row. All results are computed on the whole test set for one run. We highlight in bold the best performance for each configuration.

| Models | Gaussian | | | Radon | | |
|---|---|---|---|---|---|---|
| | 20 | 50 | 100 | 5 | 10 | 20 |
| AE-R | $.49 \pm .02$ | $\mathbf{.64 \pm .02}$ | $\mathbf{.73 \pm .01}$ | $.58 \pm .02$ | $.69 \pm .01$ | $.77 \pm .01$ |
| AE-P | $.49 \pm .02$ | $.42 \pm .02$ | $.40 \pm .02$ | $.66 \pm .01$ | $.47 \pm .02$ | $.43 \pm .02$ |
| AE-E2E | $\mathbf{.62 \pm .02}$ | $.59 \pm .02$ | $.60 \pm .02$ | $\mathbf{.83 \pm .01}$ | $\mathbf{.84 \pm .01}$ | $\mathbf{.85 \pm .01}$ |

From the results, we can see how AE-E2E outperforms the other methods in most cases. However, while in Radon longer trajectories lead to higher performance, that is not the case for Gaussian measurements, where increasing the number of measurements leads to worse performance. Note, however, that this is the case only for the adaptive strategies, while AE-R keeps improving performance the more we add measurements and observations. We conjecture that this behavior could be related to the high dimensionality of the action space for the policy with Gaussian measurements. However, the adaptive E2E model still performs better for trajectories of length $\sim 40$, which is more than is used in papers such as Bakker et al. (2020); Yin et al. (2021). As we see that generally, the worst-case error improves as the mean performance improves, we drop this metric in the following sections. We also drop the comparison with AE-P, as in our experiments, it never outperforms AE-E2E.

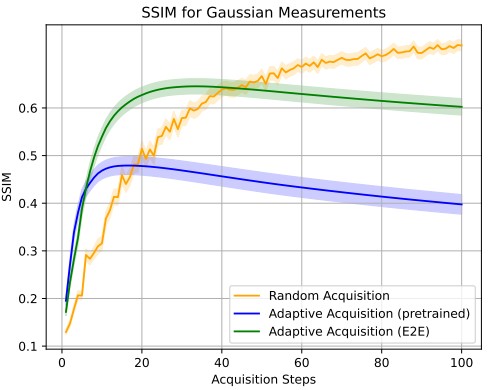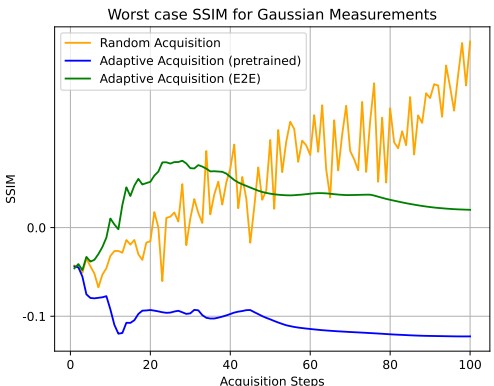

Figure 3: Results on the MNIST test dataset with Gaussian measurements. We report the mean and standard error of the mean in SSIM (Left) and worst case error in SSIM (Right) for AE-R (yellow), AE-P (blue), and AE-E2E (green) for each acquisition step in the trajectory. Each model is trained on optimizing the whole trajectory length (100 for Gaussian).

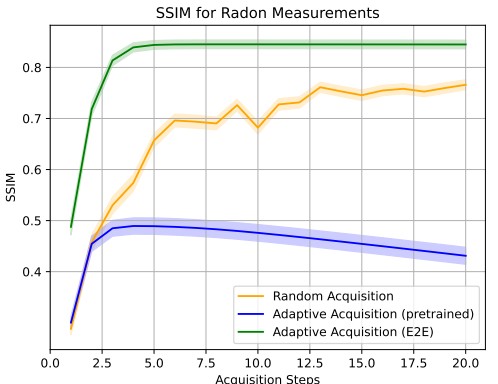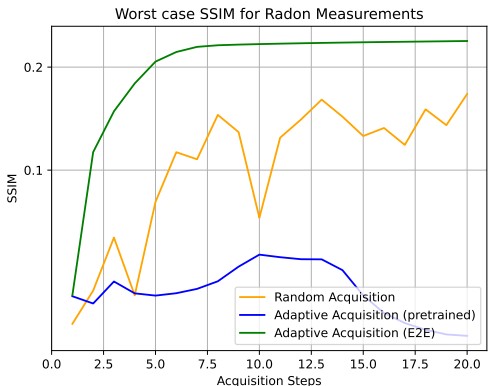

Figure 4: Results on the MNIST test dataset with Radon measurements. We report the mean and standard error of the mean in SSIM (Left) and worst case error in SSIM (Right) for AE-R (yellow), AE-P (blue), and AE-E2E (green) for each acquisition step in the trajectory. Each model is trained on optimizing the whole trajectory length (20 for Radon).

## 4.4 VAEs and $\beta$-VAEs

We perform the same experiments on MNIST done in section 4.3 but with the variational formulation introduced in section 2.5. We define the models as VAE-R and VAE-E2E. We further define two variations of VAE-E2E, in which we introduce a weighting of the KL term with a scalar parameter $\beta$, as done in Higgins et al. (2017). Using $\beta$ in a $\beta$-VAE can control the disentanglement of the features in the latent representation. A high value ($\beta > 1$) corresponds to high disentanglement, while a low value ($\beta < 1$) usually results in a better reconstruction quality for the decoder. It is, therefore, crucial for us to tune $\beta$, as a disentangled latent space can be useful for the policy, but at the same time can harm the reconstruction performance for the decoder, which is the most important metric in our setting. We try three values of $\beta : 1, 0.1$, and $0.01$, without hyperparameter tuning. The results are reported in table 2 and Figure 5.

VAE-E2E with $\beta = 0.01$ outperforms the other VAE-E2E versions and achieves performance on-par or superior compared to the equivalent AE-E2E. Furthermore, by looking at Figure 5, it is possible to see how the decrease in performance with long acquisition horizons seems to be ameliorated. This suggests that the disentanglement in the latent space with the probabilistic formulation if carefully tuned, can be useful for

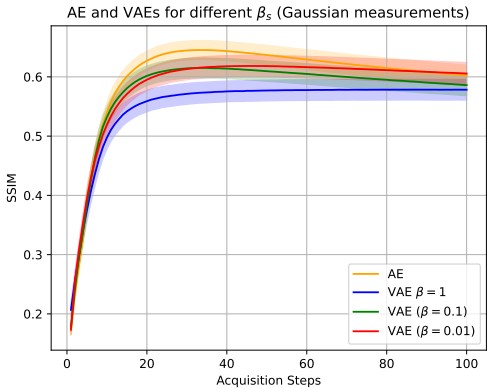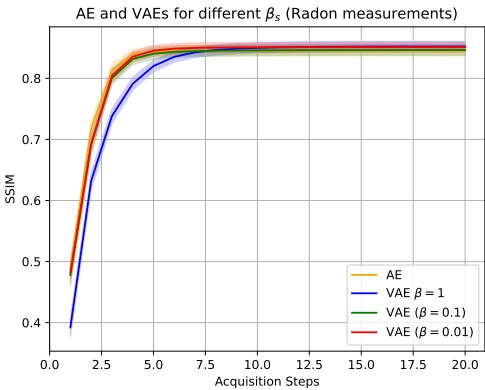

Figure 5: Comparison of AE-E2E (yellow) and VAE-E2E for different $\beta$ ($\beta = 1 \rightarrow$ blue, $\beta = 0.1 \rightarrow$ green, $\beta = 0.01 \rightarrow$ red). We show the mean and standard error of the mean in SSIM for the MNIST test set, at different stages of the acquisition trajectory. We test on models trained on different acquisition horizons: 100 for Gaussian and 20 for Radon.

the policy in selecting optimal actions in long trajectories. However, VAE-R still outperforms the adaptive strategy on long acquisition trajectories.

Table 2: MNIST SSIM (the higher the better) for different trajectory lengths (number on second row). SSIM is computed at the end of each acquisition trajectory, and we report mean and standard error of the mean. We highlight in bold the best performance for each configuration. Note that for 50 Gaussian measurements, the performances of VAE-E2E and VAE-R agree within the error.

|          |          | Gaussian |          |          | Radon     |            |           |
|----------|----------|----------|----------|----------|-----------|------------|-----------|
| Models   | $\beta$  | 20       | 50       | 100      | 5         | 10         | 20        |
| VAE-R    | 1        | $.44 \pm .02$ | $\mathbf{.63 \pm 0.02}$ | $\mathbf{.70 \pm .01}$ | $.55 \pm .02$ | $.68 \pm .013$ | $.74 \pm .01$ |
| VAE-E2E  | 1        | $.57 \pm .02$ | $.60 \pm .02$ | $.58 \pm .02$ | $.77 \pm .01$ | $.83 \pm .01$ | $.85 \pm .01$ |
|          | 0.1      | $.59 \pm .02$ | $.58 \pm .02$ | $.59 \pm .02$ | $.81 \pm .01$ | $.84 \pm .01$ | $.85 \pm .01$ |
|          | 0.01     | $\mathbf{.61 \pm .02}$ | $.63 \pm .02$ | $.61 \pm .02$ | $\mathbf{.82 \pm .01}$ | $\mathbf{.85 \pm .01}$ | $\mathbf{.85 \pm .01}$ |

### 4.5 High-resolution experiments on MAYO

Table 3: Results on the MAYO datset for both Gaussian and Radon measurements, for trajectory length respectively of 50 and 10. We report mean and standard error of the mean in SSIM on the test set. We highlight in bold the best performance for each configuration.

| Models | Gaussian 50 | Radon 10 |
|--------|-------------|----------|
| AE-R   | $\mathbf{.575 \pm .008}$ | $.444 \pm .009$ |
| AE-E2E | $.506 \pm .008$ | $\mathbf{.623 \pm .012}$ |
| VAE-R ($\beta = 1$) | $.521 \pm .008$ | $.551 \pm .010$ |
| VAE-E2E ($\beta = 1$) | $.413 \pm .012$ | $.608 \pm .011$ |

Finally, we test our methods on the higher-resolution images from the MAYO dataset. We train the models AE-R, AE-E2E, and the corresponding variational models with $\beta = 1$. We test both Gaussian and Radon measurements, with $T = 50$ and $T = 10$ respectively. The results are reported in Table 3 and Figure 6. For

Gaussian measurements, AE-R and VAE-R outperform their adaptive counterparts. We make the hypothesis that the way we use the policy is inefficient for Gaussian measurements, as the dimensionality of the action space scales quadratically with the image dimension (the action space is a vector of dimensions $1 \times 16384$). For Radon, while the adaptive models still outperform the random baselines, we observe a small decrease in performance over the trajectory for AE-E2E, which could be caused by overfitting in the unstructured latent space.

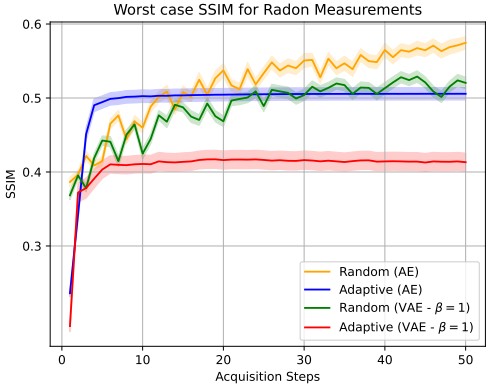 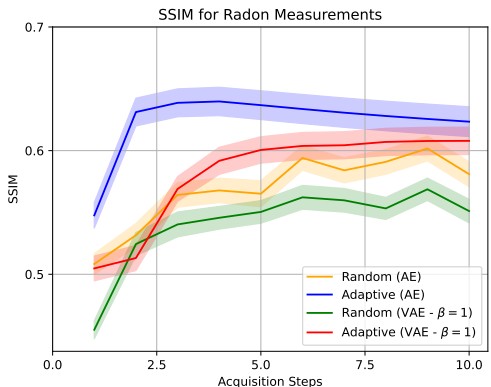

Figure 6: Results on the MAYO test dataset, for AE-R (yellow), VAE-R ($\beta = 1$, green), AE-E2E (blue) and VAE-E2E ($\beta = 1$, red), as the mean and standard error of the mean in SSIM. Left: Gaussian measurements with trajectory length 50. Right: Radon measurements with trajectory length 10.

## 5 Conclusion

We introduce a novel framework for end-to-end training of reconstruction and acquisition in generic compressed sensing problems. We show how using adaptive acquisition strategies can improve over random measurements, especially for a limited number of acquisition steps. We further introduced a variational formulation to obtain a better structure for the latent space, which when carefully tuned, can outperform the non-variational counterpart. Finally, we provided an ablation study over the effect of the choice of discount factor and policy gradient algorithm.

**Future work.** Our method does not outperform the random baseline in the cases of long acquisition horizons and high-dimensional action space. This is expected as random measurements are known to be theoretically optimal when a sufficiently long measurement horizon is available. Nonetheless, we propose some future directions to potentially improve upon our results. A careful tuning of discount factor and other policy parameters can improve performances (see App. D.2). For the case of Gaussian measurements, the dimensionality of the action space scales quadratically with the dimension of the images. Results suggest that this might be a major drawback, as the policy seems to find it difficult to select optimal actions in such a vast action space. Better ways to parametrize the probabilistic policy space could substantially improve the results, alongside more sophisticated policy learning strategies that can deal with such a complex search space. Finally, for the models using the variational formulation, a fine-grained tuning of $\beta$ could also lead to superior results, finding the optimal trade-off between latent space disentanglement and reconstruction quality.

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

# A  Gelfand Width Bounds for Adaptive Acquisition

In this section, we consider the argument based on Gelfand width analysis against the gain of adaptive sensing. In the classical result, the focus has been on the worst-case error for deterministic recovery algorithms and adaptive schemes that rely only on the outcome of previous measurements and not on the intermediate reconstruction. We extend the existing results and show that a similar result can be obtained even if we extend the adaptive schemes to use previous reconstructions as input or fix the recovery algorithm. This means that we might not expect additional gain by using previous reconstructions. We argue for two insights from the theory. First, the gain can show itself if we move away from worst-case error analysis. Second, we argue that the theory does not apply to a probabilistic adaptive scheme. This is a motivation behind using probabilistic formulations as presented above to get gains from adaptive sensing.

We follow the notation used in Foucart and Rauhut (2013) and Gelfand width-based analysis as in Foucart et al. (2010). To this end, in this section $m$ refers to the rows of the sensing matrix $\boldsymbol{A}$, which in the rest of the text is referred to as $T$. Note that in this work, we have focused on noiseless observations, convenient for Gelfand width-based analysis.

## A.1  A Classical Result

We start with some definitions and state the classical result mainly taken from Chapter 10 of Foucart and Rauhut (2013), which appeared already in Cohen et al. (2009). The first definition will provide the best possible *worst* case error that we can get over all possible $m$-dimensional non-adaptive linear measurements, denoted by $\boldsymbol{A}$, and recovery algorithms $\Delta$.

**Definition A.1.** The compressive $m$-width of a subset $K$ of a normed space $X$ is defined as:

$$E^m(K, X) := \inf_{\boldsymbol{A}, \Delta} \left( \sup_{\boldsymbol{x} \in K} \|\boldsymbol{x} - \Delta(\boldsymbol{A}\boldsymbol{x})\| , \boldsymbol{A} : X \to \mathbb{R}^m, \boldsymbol{A} \text{ is linear }, \Delta : \mathbb{R}^m \to X \right).$$

The next definition extends the previous one to the case of adaptive measurements.

**Definition A.2.** The adaptive compressive $m$-width of a subset $K$ of a normed space $X$ is defined as:

$$E_{\mathrm{ada}}^m(K, X) := \inf_{\boldsymbol{F}, \Delta} \left( \sup_{\boldsymbol{x} \in K} \|\boldsymbol{x} - \Delta(\boldsymbol{F}\boldsymbol{x})\| , \boldsymbol{F} : X \to \mathbb{R}^m, \boldsymbol{F} \text{ is adaptive }, \Delta : \mathbb{R}^m \to X \right).$$

Let's clarify what we mean by adaptive measurement. Consider the first measurement given by a linear functional $\lambda_1(\boldsymbol{x})$. The adaptive map is defined as

$$\boldsymbol{F} = \begin{pmatrix} \lambda_1(\boldsymbol{x}) \\ \lambda_{2;\lambda_1(\boldsymbol{x})}(\boldsymbol{x}) \\ \vdots \\ \lambda_{m;\lambda_1(\boldsymbol{x}),\lambda_{2;\lambda_1(\boldsymbol{x})}(\boldsymbol{x}),\dots,\lambda_{m-1;\dots,\lambda_1(\boldsymbol{x})}(\boldsymbol{x})}(\boldsymbol{x}) \end{pmatrix}$$

This simply means that the current measurement depends on the outcome of all previous measurements.

It is clear that optimal adaptive measurements, as defined here, can at least match the performance of linear measurements. This implies that $E_{\mathrm{ada}}^m(K, X) \leq E^m(K, X)$. The next question is to see if adaptive measurements can bring additional gains. To pinpoint the result, we need to introduce the notion of Gelfand $m$-width.

**Definition A.3.** The Gelfand $m$-width of a subset $K$ of a normed space $X$ is defined as

$$d^m(K, X) := \inf \left( \sup_{\boldsymbol{x} \in K \cap \ker(\boldsymbol{A})} \|\boldsymbol{x}\| , \boldsymbol{A} : X \to \mathbb{R}^m, \boldsymbol{A} \text{ is linear} \right).$$

The Gelfand width provides a common ground for comparing adaptive and non-adaptive compressed sensing widths.

**Theorem A.4** (Theorem 10.4. (Foucart and Rauhut, 2013))**.** *If $K$ is a subset of a normed space $X$, it is symmetric $K = -K$, and satisfies $K + K \subset aK$ for a positive constant $a > 0$, we have:*

$$d^m(K, X) \le E^m_{ada}(K, X) \le E^m(K, X) \le a.d^m(K, X).$$

This theorem already implies that adaptive and non-adaptive best worst-case errors are constrained from both directions by the Gelfand width. For example, in sparse recovery problems, the number of required measurements for recovering a $s$-sparse vector scales as $\Omega(s \log(N/s))$. The theorem implies that one cannot hope for better scaling with adaptive measurements. Apparently, many experimental results seem to counter this conclusion and show a concrete gain for adaptive measurements.

We would like to characterize the reason behind this discrepancy by examining some conjectures around it.

- Theorem A.4 seems to be about *the worst case error* only. The other statistics of the error can be potentially improved by adaptiveness. This hypothesis is stated in Foucart and Rauhut (2013).

- The notion of adaptiveness is too restrictive. The adaptive $\boldsymbol{F}$ only depends on the outcome of previous measurements and not necessarily the reconstructed vector from those measurements or the residual errors. These are commonly used in the literature for building adaptive methods.

- The infimum computed over all possible linear measurements and recovery algorithms assumes an exhaustive search rarely done in practice. For a suboptimal recovery method, adaptive measurements can provide gain.

These conjectures are formulated based on inspection of the definitions. In what follows, we carefully examine these conjectures. Before that, we look into the proof of the theorem again following what was presented in Foucart and Rauhut (2013).

## A.2 Proof Outline

**Proof of $d^m(K, X) \le E^m_{\text{ada}}(K, X)$.**

Consider any reconstruction map $\Delta(\cdot)$ and an adaptive sensing matrix $\boldsymbol{F}$. The main idea behind this inequality is to construct a non-adaptive linear transformation $\boldsymbol{A}$ from $\boldsymbol{F}$. To do so, consider vectors in $K$ that are in the kernel of the first measurement, namely all $\boldsymbol{x} \in K$ such that $\lambda_1(\boldsymbol{x}) = 0$. From this set, select all the vectors in the kernel of the second adaptive measurement, which is:

$$\left\{ \boldsymbol{x} \in K \cap \ker(\lambda_1) : \lambda_{2;\lambda_1(\boldsymbol{x})}(\boldsymbol{x}) = \lambda_{2;0}(\boldsymbol{x}) = 0 \right\}$$

Note that this set is given by $K \cap \ker(\lambda_1) \cap \ker(\lambda_{2;0})$. Continuing this process successively, we arrive at the set of vectors $\boldsymbol{x}$ in the set $K \cap \ker(\lambda_1) \cap \ker(\lambda_{2;0}) \cap \cdots \cap \ker(\lambda_{m;0,\ldots,0})$, for which $\boldsymbol{F}(\boldsymbol{x}) = 0$. Define $\boldsymbol{A}$ as:

$$\boldsymbol{A} = \begin{pmatrix} \lambda_1(\boldsymbol{x}) \\ \lambda_{2;0}(\boldsymbol{x}) \\ \vdots \\ \lambda_{m;0,\ldots,0}(\boldsymbol{x}) \end{pmatrix}. \tag{13}$$

Note that for all $\boldsymbol{x} \in K \cap \ker(\boldsymbol{A})$, we have $\boldsymbol{F}(\boldsymbol{x}) = 0$. Using symmetry of $K$, we get $-\boldsymbol{x} \in K \cap \ker(\boldsymbol{A})$ and therefore $\boldsymbol{F}(-\boldsymbol{x}) = 0$. We can now use the triangle inequality to get for all $\boldsymbol{x} \in K \cap \ker(\boldsymbol{A})$:

$$\|\boldsymbol{x}\|_2 = \left\| \frac{1}{2}(\boldsymbol{x} - \Delta(\boldsymbol{F}(\boldsymbol{x}))) - \frac{1}{2}(-\boldsymbol{x} - \Delta(\boldsymbol{F}(-\boldsymbol{x}))) \right\|_2 \tag{14}$$

$$\le \frac{1}{2} \|\boldsymbol{x} - \Delta(\boldsymbol{F}(\boldsymbol{x}))\|_2 + \frac{1}{2} \|-\boldsymbol{x} - \Delta(\boldsymbol{F}(-\boldsymbol{x}))\|_2 \tag{15}$$

$$\le \sup_{\boldsymbol{x} \in K} \|\boldsymbol{x} - \Delta(\boldsymbol{F}\boldsymbol{x})\|_2. \tag{16}$$

That's how the Gelfand width is connected to adaptive compressive width. We have:

$$d^m(K, X) \leq \sup_{\boldsymbol{x} \in K \cap \ker(\boldsymbol{A})} \|\boldsymbol{x}\|_2 \tag{17}$$

$$\leq \sup_{\boldsymbol{x} \in K} \|\boldsymbol{x} - \Delta(\boldsymbol{F}\boldsymbol{x})\|_2. \tag{18}$$

Since this holds for any $\Delta$ and $\boldsymbol{F}$, we can get the infimum over them and obtain:

$$d^m(K, X) \leq E_{\mathrm{ada}}^m(K, X).$$

Note that the key elements of the proof are first building the matrix $\boldsymbol{A}$ from $\boldsymbol{F}$, and having $\Delta(\boldsymbol{F}(\boldsymbol{x})) = \Delta(\boldsymbol{F}(-\boldsymbol{x}))$. The rest of the operation holds regardless.

**Proof of $E^m(K, X) \leq ad^m(K, X)$.**

The starting point is this inequality, which holds in general for any $\boldsymbol{A}$ and $\Delta$:

$$E^m(K, X) \leq \sup_{\boldsymbol{x} \in K} \|\boldsymbol{x} - \Delta(\boldsymbol{A}\boldsymbol{x})\|_2.$$

To get to Gelfand width at the output, we must select $\Delta$ properly. Interestingly, the only condition required for $\Delta$ is consistency. The condition states that for any $\boldsymbol{x}$ in $K$ and the observation $\boldsymbol{y} = \boldsymbol{A}\boldsymbol{x}$, the recovery algorithm $\Delta$ returns a vector $\hat{\boldsymbol{x}}$ that is in $K$ and in the pre-image of $\boldsymbol{y}$, $\boldsymbol{A}^{-1}(\boldsymbol{y})$, namely $\boldsymbol{A}\hat{\boldsymbol{x}} = \boldsymbol{A}\boldsymbol{x}$:

$$\Delta(\boldsymbol{y}) \in K \cap \boldsymbol{A}^{-1}(\boldsymbol{y}).$$

With this mild assumption, we get:

$$\|\boldsymbol{x} - \Delta(\boldsymbol{A}\boldsymbol{x})\|_2 \leq \sup_{\boldsymbol{z} \in K \cap \boldsymbol{A}^{-1}(\boldsymbol{A}\boldsymbol{x})} \|\boldsymbol{x} - \boldsymbol{z}\|_2$$

and thereby:

$$\sup_{\boldsymbol{x} \in K} \|\boldsymbol{x} - \Delta(\boldsymbol{A}\boldsymbol{x})\|_2 \leq \sup_{\boldsymbol{x} \in K} \sup_{\boldsymbol{z} \in K \cap \boldsymbol{A}^{-1}(\boldsymbol{A}\boldsymbol{x})} \|\boldsymbol{x} - \boldsymbol{z}\|_2.$$

Note that $\boldsymbol{x} - \boldsymbol{z} \in \ker(\boldsymbol{A})$, and $\boldsymbol{x} - \boldsymbol{z} \in K - K \subset aK$, and therefore:

$$\sup_{\boldsymbol{x} \in K} \|\boldsymbol{x} - \Delta(\boldsymbol{A}\boldsymbol{x})\|_2 \leq \sup_{\boldsymbol{x} \in K} \sup_{\boldsymbol{z} \in K \cap \boldsymbol{A}^{-1}(\boldsymbol{A}\boldsymbol{x})} \|\boldsymbol{x} - \boldsymbol{z}\|_2 \leq \sup_{\boldsymbol{x} \in aK \cap \ker(\boldsymbol{A})} \|\boldsymbol{x}\|_2 = a \sup_{\boldsymbol{x} \in K \cap \ker(\boldsymbol{A})} \|\boldsymbol{x}\|_2.$$

The rest of the proof is about taking the infimum over $\boldsymbol{A}$ and $\Delta$ from both sides, which gives us $E^m(K, X) \leq ad^m(K, X)$. In this part, the key element of the proof was the consistency assumption for $\Delta$.

We are now ready to explore if the theoretical results hold by relaxing some of the assumptions.

### A.3 *More* inputs for adaptive measurements will not help

As we mentioned above, one of the conjectures was the limited notion of adaptiveness used in the result. For example, in Bakker et al. (2020), the input to the policy network is the reconstruction from the previous measurements.

We extend the definition of adaptive sensing to include previous measurements and construction. This would include all the information available in the reconstruction pipeline apart from recovery algorithm details.

We extend the definition of the recovery algorithm $\Delta$ to incorporate intermediate reconstruction:

$$\Delta := \left\{ \Delta_j, j \in [m], \Delta_j : \mathbb{R}^j \to X \right\}. \tag{19}$$

Consider again the first measurement $\lambda_1(\boldsymbol{x})$. In the extended setting, the next measurement would be given by:

$$\lambda_{2; \lambda_1(\boldsymbol{x}), \Delta_1(\lambda_1(\boldsymbol{x}))}(\boldsymbol{x}).$$

The new adaptive map is then defined as

$$\boldsymbol{G} = \begin{pmatrix} \lambda_1(\boldsymbol{x}) \\ \lambda_{2;\lambda_1(\boldsymbol{x}),\Delta_1(\lambda_1(\boldsymbol{x}))}(\boldsymbol{x}) \\ \vdots \\ \lambda_{m;\lambda_1(\boldsymbol{x}),\Delta_1(\lambda_1(\boldsymbol{x})),\lambda_{2;\lambda_1(\boldsymbol{x}),\Delta_1(\lambda_1(\boldsymbol{x}))}(\boldsymbol{x}),\ldots,\lambda_{m-1;\ldots,\lambda_1(\boldsymbol{x}),\Delta_1(\lambda_1(\boldsymbol{x}))}(\boldsymbol{x}),\Delta_{m-1}(\lambda_{m-1;\ldots}(\boldsymbol{x}))}(\boldsymbol{x}) \end{pmatrix}$$

Just to get a better intuition from this cumbersome notation, we denote the individual measurement obtained at step $j$ by $y_j$, and the measurement vector up to time $j$ by $\boldsymbol{y}_j = (y_1, \ldots, y_j)^\top$. The extended adaptive sensing matrix is then given by:

$$\boldsymbol{G} = \begin{pmatrix} \lambda_1(\boldsymbol{x}) \\ \lambda_{2;\boldsymbol{y}_1,\Delta_1(\boldsymbol{y}_1)}(\boldsymbol{x}) \\ \lambda_{3;\boldsymbol{y}_2,\Delta_1(\boldsymbol{y}_1),\Delta_2(\boldsymbol{y}_2)}(\boldsymbol{x}) \\ \vdots \\ \lambda_{m;\boldsymbol{y}_m,\Delta_1(\boldsymbol{y}_1),\ldots,\Delta_{m-1}(\boldsymbol{y}_{m-1})}(\boldsymbol{x}). \end{pmatrix} \tag{20}$$

The question is whether the previous lower bound using Gelfand width would hold for this new adaptive sensing matrix. Let's start with the new definition.

**Definition A.5.** The extended adaptive compressive $m$-width of a subset $K$ of a normed space $X$ is defined as:

$$E^m_{\text{ext.ada}}(K, X) := \inf_{\boldsymbol{G},\Delta} \left( \sup_{\boldsymbol{x} \in K} \|\boldsymbol{x} - \Delta(\boldsymbol{G}\boldsymbol{x})\|, \boldsymbol{G} : X \to \mathbb{R}^m, \boldsymbol{G} \text{ is adaptive and given in equation 20 }, \Delta \text{ is given in equation 19} \right)$$

The following theorem states that the extended adaptive measurements do not bring any gain either.

**Theorem A.6.** *If $K$ is a subset of a normed space $X$, it is symmetric $K = -K$, and satisfies $K + K \subset aK$ for a positive constant $a > 0$, we have:*

$$d^m(K, X) \leq E^m_{ext.ada}(K, X) \leq E^m(K, X) \leq a.d^m(K, X).$$

*Proof.* The upper bounds are just replications of what we proved before. So, we only need to prove $d^m(K, X) \leq E^m_{\text{ext.ada}}(K, X)$.

First of all, see that we can build a matrix $\boldsymbol{A}$ in a similar way by considering the vectors in $K \cap \ker(\lambda_1) \cap \ker(\lambda_{2;\boldsymbol{0}_1,\Delta_1(\boldsymbol{0}_1)}) \cap \cdots \cap \ker(\lambda_{m;\boldsymbol{0}_{m-1},\Delta_1(\boldsymbol{0}_1),\ldots,\Delta_{m-1}(\boldsymbol{0}_{m-1})})$, for which $\boldsymbol{G}(\boldsymbol{x}) = 0$, namely:

$$\boldsymbol{A} = \begin{pmatrix} \lambda_1(\boldsymbol{x}) \\ \lambda_{2;\boldsymbol{0}_1,\Delta_1(\boldsymbol{0}_1)}\boldsymbol{x}) \\ \vdots \\ \lambda_{m;\boldsymbol{0}_{m-1},\Delta_1(\boldsymbol{0}_1),\ldots,\Delta_{m-1}(\boldsymbol{0}_{m-1})}(\boldsymbol{x}) \end{pmatrix}. \tag{21}$$

Now using this $\boldsymbol{A}$, we have again that for all $\boldsymbol{x} \in K \cap \ker(\boldsymbol{A})$, $\boldsymbol{G}(\boldsymbol{x}) = 0$. We can replicate the proof:

$$\|\boldsymbol{x}\|_2 = \left\| \frac{1}{2}(\boldsymbol{x} - \Delta(\boldsymbol{G}(\boldsymbol{x})) - \frac{1}{2}(-\boldsymbol{x} - \Delta(\boldsymbol{G}(-\boldsymbol{x})) \right\|_2 \tag{22}$$

$$\leq \frac{1}{2} \|\boldsymbol{x} - \Delta(\boldsymbol{G}(\boldsymbol{x}))\|_2 + \frac{1}{2} \|-\boldsymbol{x} - \Delta(\boldsymbol{G}(-\boldsymbol{x}))\|_2 \tag{23}$$

$$\leq \sup_{\boldsymbol{x} \in K} \|\boldsymbol{x} - \Delta(\boldsymbol{G}\boldsymbol{x})\|_2, \tag{24}$$

which implies:

$$d^m(K, X) \leq \sup_{\boldsymbol{x} \in K \cap \ker(\boldsymbol{A})} \|\boldsymbol{x}\|_2 \leq \sup_{\boldsymbol{x} \in K} \|\boldsymbol{x} - \Delta(\boldsymbol{G}\boldsymbol{x})\|_2 \leq E^m_{\text{ext.ada}}(K, X). \tag{25}$$

$\square$

*Remark* A.7. Note that in the above argument, the key inequality is $\sup_{\boldsymbol{x}\in K\cap\ker(\boldsymbol{A})}\|\boldsymbol{x}\|_2 \leq \sup_{\boldsymbol{x}\in K}\|\boldsymbol{x}-\Delta(\boldsymbol{Gx})\|_2$. If the choice sensing matrix, adaptive or not, is limited to a specific subset of all matrices, say $\mathcal{A}$, then we can redefine the Gelfand width limited to $\mathcal{A}$, and obtain a similar lower bound.

### A.4 Adaptive sensing would not improve for a fixed recovery algorithm

We now examine the third hypothesis, which is about fixing the recovery algorithm, which might be suboptimal.

Note that the lower bound would still hold regardless of the choice of $\Delta$, namely $d^m(K, X) \leq \sup_{\boldsymbol{x}\in K}\|\boldsymbol{x}-\Delta(\boldsymbol{Gx})\|_2$. This time, we need to examine the upper bound. It turns out that as long as the recovery algorithm is consistent, that is, $\Delta(\boldsymbol{y}) \in K \cap \boldsymbol{A}^{-1}(\boldsymbol{y})$, we still get:

$$\|\boldsymbol{x}-\Delta(\boldsymbol{Ax})\|_2 \leq \sup_{\boldsymbol{z}\in K\cap\boldsymbol{A}^{-1}(\boldsymbol{Ax})}\|\boldsymbol{x}-\boldsymbol{z}\|_2 \leq a \sup_{\boldsymbol{x}\in K\cap\ker(\boldsymbol{A})}\|\boldsymbol{x}\|_2. \tag{26}$$

The upper bound follows accordingly. This simple derivation shows that not much is to be expected from adaptive sensing, even if the recovery algorithm has limitations.

We can repeat this argument by considering a subset of all possible sensing matrices. It can be similarly shown that a similar bound can be obtained on the errors.

### A.5 Where is the gain of adaptive sensing?

As we have seen in the above derivations, some of the conjectures about the source of gain in adaptive sensing can be debunked. To summarize, even extending the notion of adaptiveness or restricting the measurement matrix and recovery algorithm sets would not break the theorem. So the natural question is where else we can look for the gains of adaptive sensing.

The most obvious angle, previously mentioned in the literature, is about moving from worst-case error to average error. This change will break some of the inequalities used in the proof, for example, $\|\boldsymbol{x}-\Delta(\boldsymbol{Ax})\|_2 \leq \sup_{\boldsymbol{z}\in K\cap\boldsymbol{A}^{-1}(\boldsymbol{Ax})}\|\boldsymbol{x}-\boldsymbol{z}\|_2$.

The other key point, crucial for both lower and upper bound, was the deterministic nature of the adaptive scheme and recovery algorithm, for example, assuming $\Delta(\boldsymbol{Ax})$ does not have stochastic components for instance, a random initialization, or the adaptive measurement $\lambda_{j;...}(\boldsymbol{x})$ is a deterministic function of past. In the latter case, one cannot find a deterministic $\boldsymbol{A}$ from $\boldsymbol{G}$ to use in the lower bound proof.

The consistency was another assumption, although in most cases, we can expect the recovery algorithm to provide an approximation close enough to the underlying data manifold and satisfy measurement consistency.

To summarize, there are two angles where the gain of adaptive sensing shows itself. First, it is about moving away from worst-case errors. Second, we should consider probabilistic adaptive schemes. Our problem formulation in the paper follows these guidelines.

## B Lower bound for reconstruction loss in Adaptive Acquisition

In this section, we derive a variational bound for the reconstruction loss in adaptive acquisition schemes. The graphical model of our scenario is represented in Figure 7. The random variable $x_{1:T}$ correspond to the reconstructed signal, $a_{1:T}$ are the measurement actions chosen adaptively, and $y_{1:T}$ are the observations.

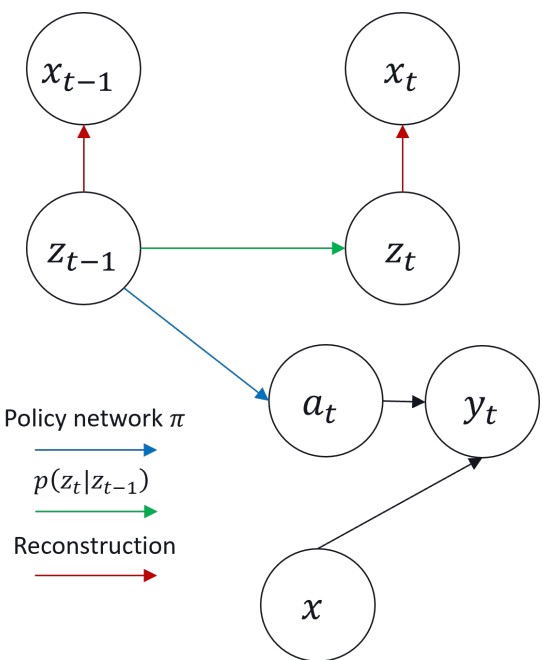

Figure 7: Graphical Model for Random Variables

The derivation runs as follows, and involves standard factorization steps with the ELBO lower bound:

$$\log p(x_{1:T} \mid a_{1:T}, y_{1:T}) = \log \int p(x_{1:T}, z_{1:T} \mid a_{1:T}, y_{1:T}) dz_{1:T} \tag{27}$$

$$= \log \int p(x_{1:T}, z_{1:T} \mid a_{1:T}, y_{1:T}) \frac{q(z_{1:T} \mid x_{1:T}, a_{1:T}, y_{1:T})}{q(z_{1:T} \mid x_{1:T}, a_{1:T}, y_{1:T})} dz_{1:T}$$

$$= \log E_{z_{1:T} \sim (z_{1:T} \mid x_{1:T}, a_{1:T}, y_{1:T})} \left[ \frac{p(x_{1:T}, z_{1:T} \mid a_{1:T}, y_{1:T})}{q(z_{1:T} \mid x_{1:T}, a_{1:T}, y_{1:T})} \right]$$

$$\overset{(a)}{\geq} E_{z_{1:T}} \left[ \log \frac{p(x_{1:T}, z_{1:T} \mid a_{1:T}, y_{1:T})}{q(z_{1:T} \mid x_{1:T}, a_{1:T}, y_{1:T})} \right]$$

$$= E_{z_{1:T}} \left[ \log \frac{\prod_{t=1}^{T} p(x_t \mid z_t, a_t, y_t) p(z_t \mid z_{1:t-1}, y_{1:t}, a_{1:t})}{\prod_{t=1}^{T} q(z_t \mid x_t, z_{1:t-1}, y_{1:t}, a_{1:t})} \right]$$

$$= E_{z_{1:T}} \left[ \sum_{t=1}^{T} \log p(x_t \mid z_t, a_t, y_t) + \log p(z_t \mid z_{1:t-1}, y_{1:t}, a_{1:t}) - \log q(z_t \mid x_t, z_{1:t-1}, y_{1:t}, a_{1:t}) \right]$$

$$= \sum_{t=1}^{T} E_{z_{1:T}} \left[ \log p(x_t \mid z_t, a_t, y_t) + \log p(z_t \mid z_{1:t-1}, y_{1:t}, a_{1:t}) - \log q(z_t \mid x_t, z_{1:t-1}, y_{1:t}, a_{1:t}) \right]$$

$$= \sum_{t=1}^{T} E_{z_T} E_{z_{1:T-1}} \left[ \log p(x_t \mid z_t, a_t, y_t) + \log p(z_t \mid z_{1:t-1}, y_{1:t}, a_{1:t}) - \log q(z_t \mid x_t, z_{1:t-1}, y_{1:t}, a_{1:t}) \right]$$

$$= \sum_{t=1}^{T} E_{z_{1:T-1}} E_{z_T} \left[ \log p(x_t \mid z_t, a_t, y_t) + \log p(z_t \mid z_{1:t-1}, y_{1:t}, a_{1:t}) - \log q(z_t \mid x_t, z_{1:t-1}, y_{1:t}, a_{1:t}) \right]$$

$$= \sum_{t=1}^{T} E_{z_{1:T-1}} E_{z_T} \left[ \log p(x_t \mid z_t, a_t, y_t) - \log \frac{q(z_t \mid x_t, z_{1:t-1}, y_{1:t}, a_{1:t})}{p(z_t \mid z_{1:t-1}, y_{1:t}, a_{1:t})} \right]$$

$$= \sum_{t=1}^{T} E_{z_{1:T-1}} \left[ E_{z_T} [\log p(x_t \mid z_t, a_t, y_t)] - E_{z_T} \left[ \log \frac{q(z_t \mid x_t, z_{1:t-1}, y_{1:t}, a_{1:t})}{p(z_t \mid z_{1:t-1}, y_{1:t}, a_{1:t})} \right] \right]$$

In the derivations, $(a)$ is the ELBO step. Note that the derivations are general and assumed full dependence on the history for the prior $p(\cdot)$ and the approximate posterior $q(\cdot)$. First, we can assume a Markov assumption on the sequence of $z_t$'s and simplify $p(z_t \mid z_{1:t-1}, y_{1:t}, a_{1:t})$ as $p(z_t \mid z_{t-1})$. Next, we assume that the latent variable $z_t$ summarizes whatever necessary for the reconstruction, namely replacing $p(x_t \mid z_t, a_t, y_t) = p(x_t \mid z_t)$. Finally, we use a recurrent architecture, a GRU, in our implementation of approximate posterior $q(\cdot)$, and there, the history of observations $y_{1:t}$, and the actions $a_{1:t}$ is summarized through a hidden state $h_t$. This means that $q(z_t \mid x_t, z_{1:t-1}, y_{1:t}, a_{1:t})$ is given by $q(z_t \mid a_t, y_t, h_t)$. Using these simplifications, we will arrive at the following equation.

$$\sum_{t=1}^{T} E_{z_{1:T-1}} \left[ E_{z_T}[\log p(x_t \mid z_t, a_t, y_t)] - E_{z_T} \left[ \log \frac{q(z_t \mid x_t, z_{1:t-1}, y_{1:t}, a_{1:t})}{p(z_t \mid z_{1:t-1}, y_{1:t}, a_{1:t})} \right] \right]$$

$$= \sum_{t=1}^{T} E_{z_{1:T-1}} [E_{z_T}[\log p(x_t = x \mid z_t, a_t, y_t)] - D_{KL}(q(z_t \mid a_t, y_t, h_t) \parallel p(z_t \mid z_{t-1}))]$$

$$= \sum_{t=1}^{T} E_{z_{1:T-1}} [E_{z_T}[\log p(x_t = x \mid z_t)] - D_{KL}(q(z_t \mid a_t, y_t, h_t) \parallel p(z_t \mid z_{t-1}))].$$

## C  Training details

For all the experiments, we train the models for 100 epochs, with batch size 128. All the models are trained with ADAM optimizer (Kingma and Ba, 2015), with learning rate $lr = 1e^{-3}$ for reconstruction and $lr = 1e^{-4}$ for acquisition. In the Gaussian case, the measurements $a_t$ are normalized before computing the corresponding observation. For Radon, after computing $y_t = R(a_t, x)$, $a_t$ is divided by $\pi$ to be in the range $[-1, 1]$, while $y_t$ is scaled to be in the range $[0, 1]$.

### C.1  Preprocessing for MAYO dataset

We use the DICOM image data consisting of $512 \times 512$ images belonging to three different classes labeled N for neuro, C for chest, and L for liver. To train our models, we consider $\sim$1.5K samples from the N subset and split them into train, validation, and test sets comprising $\sim$80%, $\sim$10%, and $\sim$10% of the images, respectively. Before feeding a model, we apply a random crop and then rescale the images to $128 \times 128$. Finally, we normalize the pixel values in $[0, 1]$.

### C.2  Architectures

**GRU encoder.**  At each time step, the vectors $a_t$ and $y_t$ are concatenated and fed to a GRU. For MNIST, the GRU has 1 layer with 128 units, while for mayo $x$ layers with $y$ units. Then, a fully connected layer maps the output of the GRU to the latent size dimension ($\times 2$ in the variational experiments to account for mean and standard deviation).
**Convolutional Decoder.**  For MNIST experiments, the decoder is a two-layer transposed convolution network with 64 and 128 channels. For MAYO, the network has 8 transposed convolution residual blocks. We do not use any normalization layer.
**Policy Network.**  The policy network has the same architecture as the Decoder when we use Gaussian measurements (outputs two channels instead of one). For Radon measurements, the policy is an MLP with one hidden layer with 256 units.

## D  Additional experimental results

### D.1  Full Results from MNIST

In Table 4 we report the full set of results concerning the MNIST dataset. Specifically, compared to Table 1, in this section, we add to the table the results from the worst-case scenario for each model we trained.

Table 4: Results on the MNIST datset for both Gaussian and Radon measurements in SSIM (higer is better). The trajectory length of the experiment is reported on the second row. For each model, we report mean and standard error of the mean on on the row signed as M, while we report the worst case error on the row signed as W. All results are computed on the whole test set for one run. We highlight in bold the best performance for each configuration.

| Models | | Gaussian | | | Radon | | |
|---|---|---|---|---|---|---|---|
| | | 20 | 50 | 100 | 5 | 10 | 20 |
| AE-R | M | $.49 \pm .02$ | $\mathbf{.64 \pm .02}$ | $\mathbf{.73 \pm .01}$ | $.58 \pm .02$ | $.69 \pm .01$ | $.77 \pm .01$ |
| | W | $-.01$ | $\mathbf{.07}$ | $\mathbf{.21}$ | $-.01$ | $.08$ | $.17$ |
| AE-P | M | $.49 \pm .02$ | $.42 \pm .02$ | $.40 \pm .02$ | $.66 \pm .01$ | $.47 \pm .02$ | $.43 \pm .02$ |
| | W | $-.12$ | $-.10$ | $-.12$ | $.03$ | $-.10$ | $-.06$ |
| AE-E2E | M | $\mathbf{.62 \pm .02}$ | $.59 \pm .02$ | $.60 \pm .02$ | $\mathbf{.83 \pm .01}$ | $\mathbf{.84 \pm .01}$ | $\mathbf{.85 \pm .01}$ |
| | W | $\mathbf{.01}$ | $-.05$ | $.02$ | $\mathbf{.22}$ | $\mathbf{.26}$ | $\mathbf{.23}$ |

## D.2 The effect of the discount factor $\gamma$

Bakker et al. (2020) suggests that greedy policies can perform on par or even outperform policies trained with a discounted reward. We investigate the role of the discount factor in this section. We additionally explore the effectiveness of the Vanilla Policy gradient and compare it to the more recent and efficient Proximal Policy Optimization (PPO) (Schulman et al., 2017). The results are reported in table 5 and Figure 8 for AE-E2E, with 20 Gaussian measurements on MNIST. Our experiments show that, at least with our model and training strategy, a carefully discounted objective function leads to the best results, as it generally happens in RL. The fact that PPO with $\gamma = 0.9$ obtains the best performance suggests that using more recent and advanced policy gradient algorithms could also bring additional improvement.

Table 5: Comparison between PPO and VPG for different discount factors. All the models are trained with trajectories of 20 Gaussian acquisitions on the MNIST test dataset. We report mean and standard error of the mean at the end of the trajectory, in SSIM (higher is better). We highlight in bold the best performance across the different configurations.

| Algorithm | Gaussian - 20 | | | |
|---|---|---|---|---|
| | $\gamma = 0$ | $\gamma = 0.9$ | $\gamma = 0.99$ | $\gamma = 1$ |
| VPG | $.577 \pm .017$ | $.616 \pm .017$ | $.634 \pm .016$ | $.618 \pm .017$ |
| PPO | $.570 \pm .017$ | $\mathbf{.641 \pm .016}$ | $.627 \pm .016$ | $.622 \pm .017$ |

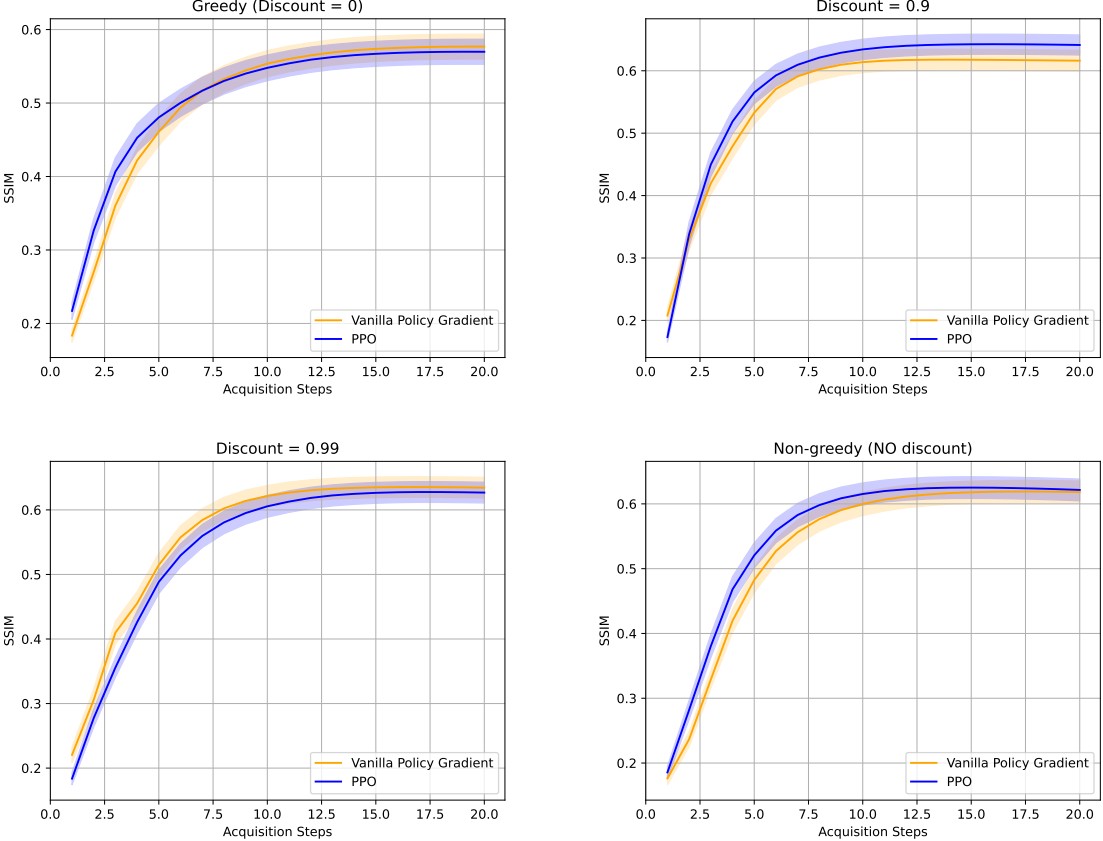

Figure 8: Mean and standard error of the mean in SSIM at each acquisition step, for models trained on MNIST with Gaussian measurements and 20 step trajectories. The results are obtained on the test set. We compare VPG (yellow) and PPO (blue) performance for different discount factors $\gamma$ reported on top of each graph.

### D.3  Comparison with ISTA

As an additional baseline, we compare the performance of our approach (AE-E2E and AE-R) with the well-established compressed sensing method Iterative Soft-Tresholding Algorithm (ISTA) (Daubechies et al., 2004) on the MNIST dataset. Both AE-E2E and AE-R are trained on a trajectory length of 784 measurements. The results are reported in figure 9.

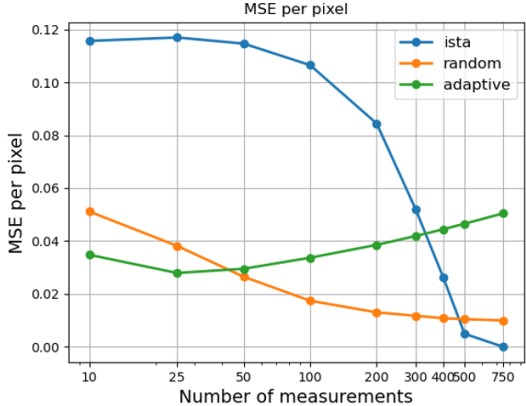

Figure 9: Performance of ISTA (blue), AE-R (orange) and AE-E2E (green) in average mean square error per pixel over the test dataset.

### D.4  Additional figures

In this section, we show additional plots for the experiments in sections 4.3 and 4.4. Figure 10 reports the results at each step of the acquisition trajectory for models trained on 20 and 50 steps for Gaussian measurements, and 5 and 10 steps for Radon, for the AE-R, AE-P, and AE-E2E models. The same is reported in Figure 11 but for the variational encoder-decoder models.

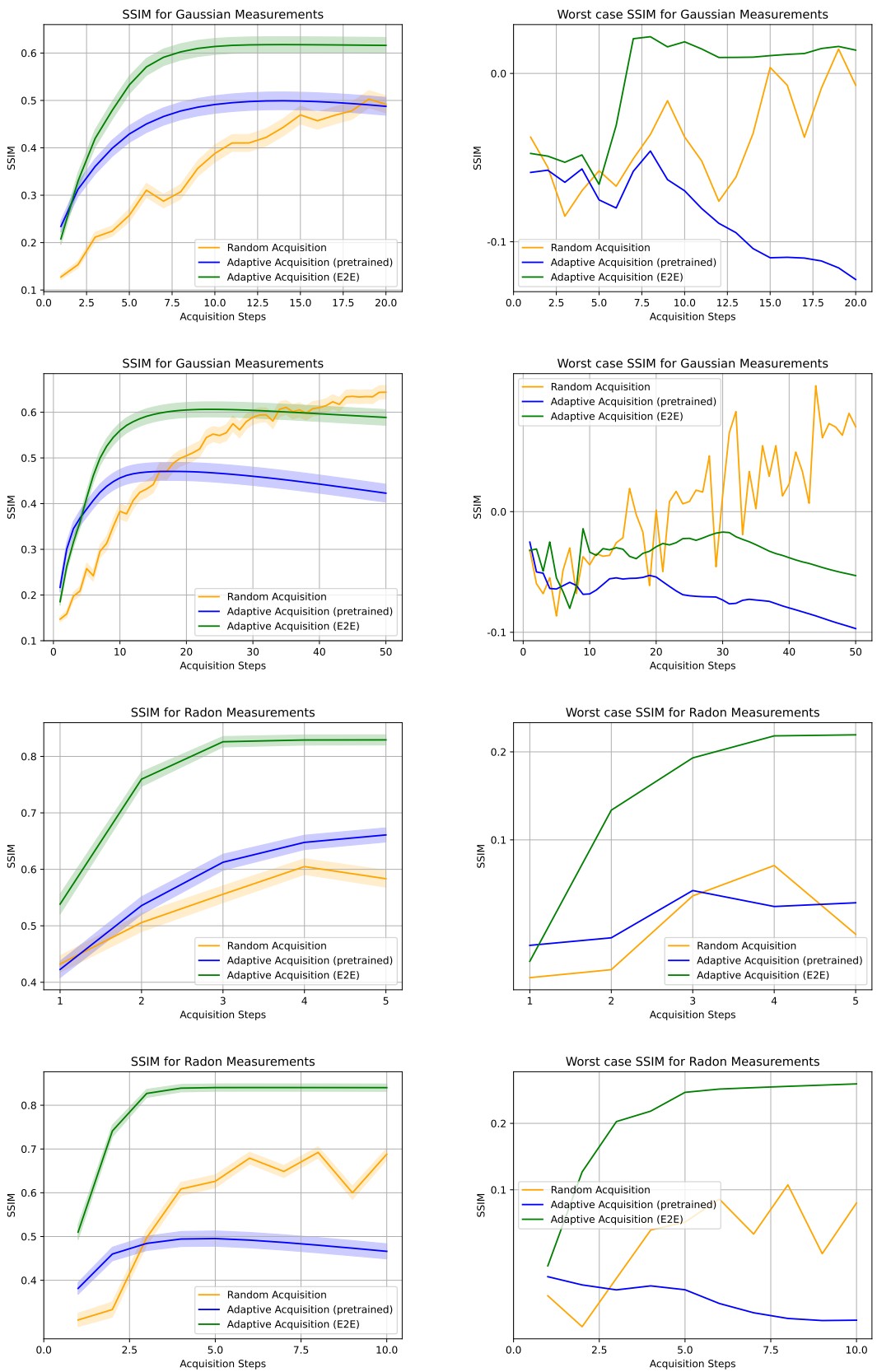

Figure 10: Results on the MNIST test dataset with Gaussian and Radon measurements. We report the mean and standard error of the mean in SSIM (Left) and worst case error in SSIM (Right) for AE-R (yellow), AE-P (blue), and AE-E2E (green) for each acquisition step in the trajectory. Each model is trained on different trajectory lengths: 20 and 50 for Gaussian and 5 and 10 for Radon.

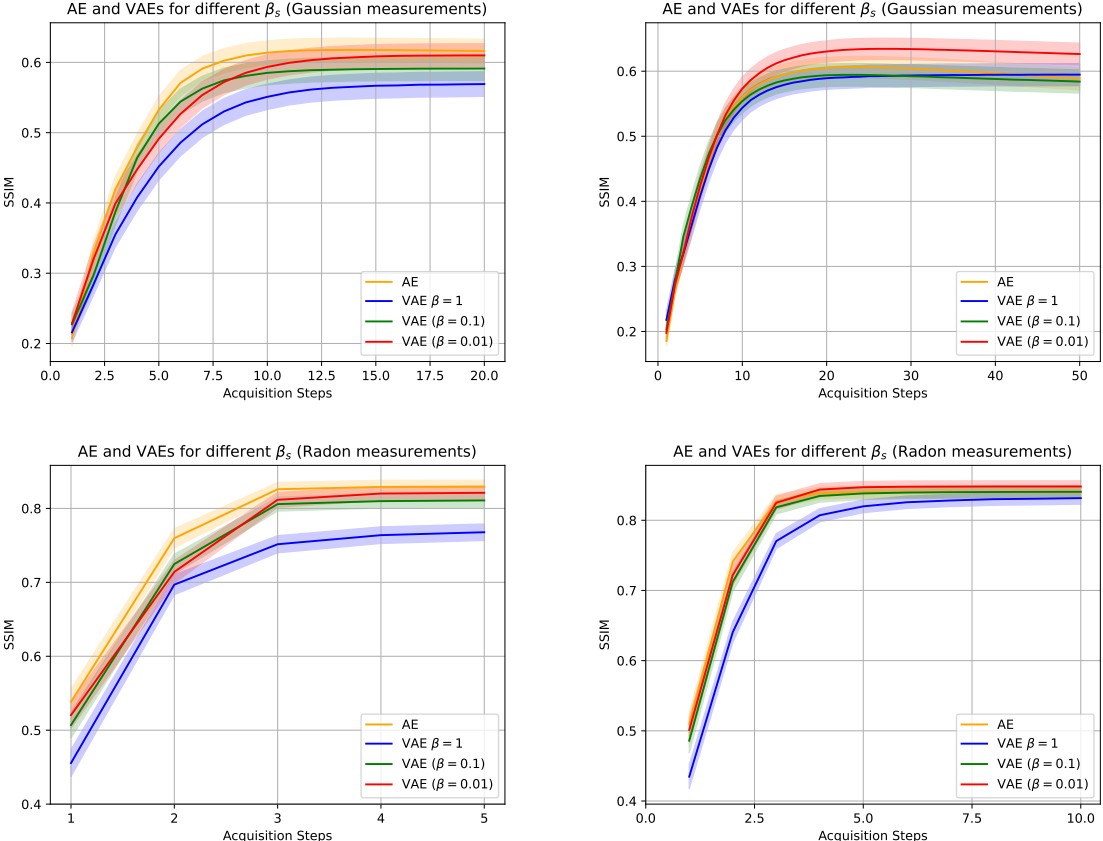

Figure 11: Comparison of AE-E2E (yellow) and VAE-E2E for different $\beta$ ($\beta = 1 \rightarrow$ blue, $\beta = 0.1 \rightarrow$ green, $\beta = 0.01 \rightarrow$ red). We show the mean and standard error of the mean in SSIM for the MNIST test set at different stages of the acquisition trajectory. We test on models trained with Gaussian measurements on 20 and 50 acquisition horizons and 5 and 10 for Radon.

### D.5 Ablation Studies

The purpose of this section is to report results from ablation studies. We report results concerning the MNIST and MAYO datasets. According to the main paper, also in this section, we consider Gaussian and Radon measurements. As we already mentioned in subsection 4.1, we base our choice of Gaussian and Radon type of measurements on the following observation. In compressed sensing, Gaussian measurements can achieve theoretical limits for non-adaptive sensing and therefore represent the best non-adaptive sensing scheme. Instead, the Radon transform represents a common choice to reconstruct images from CT scans (see Beatty (2012) for a detailed description). In Table 6 to Table 9 we report results concerning models trained using the final reward only, i.e., we consider only the reconstruction error at the final state (after $T$ measurements). We rerun the simulations on the MNIST dataset multiple times. It is possible to notice how in Table 6 and Table 7 the results are significantly worse than the cases in which reward is given at each time step, indicating that the increased sparsity in the reward distribution results to be more challenging for the policy, especially as the acquisition horizon increases. A different result can be seen in Table 8 and Table 9, where the performances seem to improve with respect to the results in Table 3, even though not always with increased number of acquisition. From these results, we can conclude that the two different kinds of rewards may be suited for different models and datasets and that both should be tested.

Table 6: Results on the MNIST dataset for both Gaussian and Radon measurements in SSIM (higer is better). The trajectory length of the experiment is reported on the second row. For each model, we report mean and standard error of the mean on the row signed as M, while we report the worst case error on the row signed as W. The reward for the RL policy is computed based on the reconstruction error at the final state only.

| Models | | Gaussian | | | Radon | | |
|--------|---|------------------|------------------|------------------|------------------|------------------|------------------|
| | | 20 | 50 | 100 | 5 | 10 | 20 |
| AE-R | M | $.505 \pm .018$ | $.641 \pm .014$ | $.726 \pm .011$ | $.658 \pm .014$ | $.664 \pm .014$ | $.757 \pm .011$ |
| | W | $-.028$ | $.064$ | $.132$ | $.074$ | $-.036$ | $.100$ |
| AE-P | M | $.325 \pm .021$ | $.217 \pm .018$ | $.232 \pm .016$ | $.680 \pm .013$ | $.456 \pm .019$ | $.365 \pm .017$ |
| | W | $-.117$ | $-.153$ | $-.119$ | $.079$ | $-.120$ | $-.111$ |
| AE-E2E | M | $.475 \pm .021$ | $.296 \pm .021$ | $.283 \pm .021$ | $.824 \pm .009$ | $.781 \pm .014$ | $.739 \pm .015$ |
| | W | $-.078$ | $-.162$ | $-.144$ | $.205$ | $.057$ | $.010$ |

Table 7: Results on the MNIST dataset for both Gaussian and Radon measurements in SSIM (higer is better). The trajectory length of the experiment is reported on the second row. For each model, we report mean and standard error of the mean. The reward for the RL policy is computed based on the reconstruction error at the final state only.

| Models | $\beta$ | Gaussian | | | Radon | | |
|---|---|---|---|---|---|---|---|
| | | 20 | 50 | 100 | 5 | 10 | 20 |
| VAE-R | 1 | $.441 \pm .018$ | $.627 \pm .014$ | $.703 \pm .012$ | $.566 \pm .016$ | $.639 \pm .015$ | $.733 \pm .012$ |
| | .1 | $.467 \pm .017$ | $.629 \pm .015$ | $.701 \pm .012$ | $.581 \pm .015$ | $.678 \pm .012$ | $.743 \pm .011$ |
| | .01 | $.482 \pm .016$ | $.635 \pm .014$ | $.696 \pm .012$ | $.595 \pm .014$ | $.714 \pm .012$ | $.761 \pm .010$ |
| | .001 | $.479 \pm .016$ | $.635 \pm .014$ | $.706 \pm .012$ | $.612 \pm .014$ | $.675 \pm .013$ | $.764 \pm .010$ |
| VAE-P | 1 | $.335 \pm .019$ | $.297 \pm .017$ | $.252 \pm .017$ | $.585 \pm .014$ | $.662 \pm .015$ | $.586 \pm .017$ |
| | .1 | $.348 \pm .019$ | $.196 \pm .015$ | $.216 \pm .018$ | $.642 \pm .015$ | $.662 \pm .014$ | $.535 \pm .016$ |
| | .01 | $.321 \pm .018$ | $.251 \pm .020$ | $.201 \pm .018$ | $.655 \pm .013$ | $.433 \pm .016$ | $.493 \pm .017$ |
| | .001 | $.323 \pm .018$ | $.203 \pm .016$ | $.235 \pm .018$ | $.613 \pm .013$ | $.520 \pm .016$ | $.378 \pm .018$ |
| VAE-E2E | 1 | $.306 \pm .021$ | $.301 \pm .021$ | $.259 \pm .019$ | $.646 \pm .015$ | $.621 \pm .016$ | $.656 \pm .017$ |
| | .1 | $.307 \pm .020$ | $.256 \pm .019$ | $.207 \pm .015$ | $.714 \pm .014$ | $.676 \pm .015$ | $.490 \pm .022$ |
| | .01 | $.293 \pm .019$ | $.231 \pm .017$ | $.205 \pm .014$ | $.722 \pm .014$ | $.654 \pm .017$ | $.573 \pm .026$ |
| | .001 | $.295 \pm .019$ | $.205 \pm .015$ | $.215 \pm .015$ | $.771 \pm .012$ | $.737 \pm .015$ | $.637 \pm .021$ |

Table 8: Results on the MAYO dataset for Radon measurements. The trajectory length of the experiment is reported on the second row. We report mean and standard error of the mean in SSIM. For each model, we report mean and standard error of the mean on the row signed as M, while we report the worst case error on the row signed as W. The reward for the RL policy is computed based on the reconstruction error at the final state only.

| Models | | Radon | | |
|---|---|---|---|---|
| | | 5 | 10 | 20 |
| AE-R | M | $.629 \pm .014$ | $\mathbf{.646 \pm .014}$ | $.532 \pm .011$ |
| | W | $.339$ | $.349$ | $.286$ |
| AE-E2E | M | $\mathbf{.657 \pm .016}$ | $\mathbf{.660 \pm .015}$ | $\mathbf{.658 \pm .016}$ |
| | W | $.242$ | $.302$ | $.264$ |

Table 9: Results on the MAYO dataset for Radon measurements. The trajectory length of the experiment is reported on the second row. We report mean and standard error of the mean in SSIM. The reward for the RL policy is computed based on the reconstruction error at the final state only.

| Models | $\beta$ | Radon | | |
|---|---|---|---|---|
| | | 5 | 10 | 20 |
| VAE-R | 1 | $.571 \pm .012$ | $.612 \pm .013$ | $.604 \pm .013$ |
| | .1 | $.574 \pm .013$ | $.618 \pm .013$ | $.620 \pm .013$ |
| | .01 | $.599 \pm .014$ | $.614 \pm .013$ | $.593 \pm .012$ |
| | .001 | $.556 \pm .011$ | $.622 \pm .013$ | $.629 \pm .013$ |
| VAE-E2E | 1 | $.643 \pm .015$ | $.652 \pm .014$ | $\mathbf{.668 \pm .014}$ |
| | .1 | $.661 \pm .014$ | $\mathbf{.664 \pm .014}$ | $\mathbf{.659 \pm .015}$ |
| | .01 | $\mathbf{.666 \pm .014}$ | $\mathbf{.661 \pm .014}$ | $\mathbf{.661 \pm .015}$ |
| | .001 | $\mathbf{.664 \pm .014}$ | $\mathbf{.662 \pm .014}$ | $\mathbf{.671 \pm .014}$ |

