# OpenReview forum: "Reinforcement Learning of Adaptive Acquisition Policies for Inverse Problems"
_TMLR — Rejected by TMLR_

### Review · Reviewer_fgGo · 2023-12-09

**Summary Of Contributions:**

This manuscript considers the problem of jointly learning a sampling strategy for obtaining measurements of a signal (acquisition strategy), and a reconstruction strategy for reconstructing the signal from those (sparse) measurements.  The work thus falls into the area of compressed sensing, where it distinguishes (according to the paper) from most of the existing work in that it trains both the acquisition and the reconstruction strategies jointly. In addition, also a variational formulation is proposed as an additional variant. The algorithms are evaluated through numerical experiments on two standard data sets.  For "Gaussian"-type measurements, the proposed strategies are found to be beneficial over standard random measurements for low-dimensional problems and short acquisition horizons; for "Radon"-type measurements, shorter horizons are investigated, where the method yields superior performance. Conclusions and hypotheses are derived from the empirical results, which might give rise to further research in this direction.

**Audience:**

No

**Broader Impact Concerns:**

In my opinion it is fine, and does not need to be specifically addressed here.

**Claims And Evidence:**

No

**Requested Changes:**

Critical ones correspond to the above mentioned weaknesses (W1) to (W7)

**Strengths And Weaknesses:**

## Strengths

(S1) Adaptive sampling is a relevant problem in compressed sensing, and only few results have attempted to jointly train acquisition and reconstruction strategies.

(S2) The empirical results support that the proposed method is superior over standard random sampling and one competitor approach in some of the considered cases (short horizon, relatively low dimensional problems). The results support insights regarding for what horizons and dimensionality adaptive approaches can make sense.

(S3) From my understanding, an interesting contribution of this work is to consider an adaptive sensing policy that is based on a trained latent state rather than the last reconstruction. This has the potential to capture and provide more relevant information for the sampling decision. It could be nice to stress this further, e.g., with empirical results specifically illustrating this aspect.



## Weaknesses

(W1) Discussion of related work does not sufficiently support the relevance of the contributions.

The innovation over the related work, and thus the relevance of this work, is not very clearly presented.  In particular, the discussion of related work in Sec. 2.1 did not allow me to really pin point what is better / the improvement over the SOTA.  In particular:

* The authors claim that they present a "more general approach" as compared to the work on MRI imaging.  However, it is not spelled out in what way their approach is more general.
* "Gelfand widt-based analysis" needs to be explained in Sec. 2.1 if it is relevant.  As is, it is not clear what this is and how it relates to the contributions of this work.
* The simultaneous consideration for acquisition and reconstruction has been done before with supervised learning.  The authors distinguish their work especially in that they use RL instead of supervised learning.  However, what is the distinctive advantage of RL / their approach is not explained, but should be.

(W2) Relevance of the considered problem

* The authors consider the case of noise-free measurements, which is obviously simpler than the noisy case, which is considered in related works on compressed sensing (see Sec. 2.1).  If the noisy case is essentially standard, then the consideration of the noise-free problem only appears to be a significant limitation of the relevance of this work.  Is this true?  Why is the noisy case not considered?
* Similarly, only linear maps are being considered.  Is this limiting or still relevant in compressed sensing?

(W3) Problem formulation is imprecise with regards to the contributions of this paper. (Section 3)

* The problem formulation (Sec. 3.1) is not specific to this paper.  From my understanding (as non-expert in compressed sensing), the described problem is the typical problem considered in compressed sensing.  So, it does not become clear what is the particular question that this paper addresses, and what distinguishes it from prior work.  This would be important to understand in order to assess the relevance.  Please provide a succinct problem formulation, and then your proposed solution.
* Goal 1: "possible if we assume a prior about the signal structure, ..." -- What is the prior structure actually assumed in this work?
* Eq. (2) seems incorrect, or is unclear.  First, \delta is not introduced.  Presumably, the authors mean the Dirac distribution.  \delta(x_t+1 - x_t) then essentially means that x_t+1 = x_t.  This is not the same as x_t being stationary and/or the transition distribution being fixed.
  In fact, I did not understand from the description what is meant.  If the distribution is supposed to be stationary, I believe that the description is incorrect.  If the authors mean that x_t+1 = x_t, then this means that the signal is considered constant; that is, there is no (non-trivial) transition dynamics and thus it is unclear why the transition dynamics are introduced and what's the benefit of treating the problem with RL rather than static optimization.  In any case, the problem statement is imprecise and unclear here.
* Observation model (Sec. 3.3):  Because F(.) has previously been introduced as A * x, F would simply be a linear observation model. I find it very misleading that the observation model is first introduced as (i) a distribution (to allow for noise/uncertainty), and (ii) nonlinear map, to then say that only linear and noiseless observations are considered.  This is misleading IMO. If the linear map is the one considered, it should be stated like this right away.
* I found the reward unclear as well.  First, SSIM should be explained.  Then, how can the reward be evaluated as it depends on the true signal x, which is not known in practice, no?

(W4) Relevance of theoretical results unclear.

* Abstract: from the formulation in the abstract, it is unclear what type of results are obtained from the mentioned "theoretical analysis". If there are clear theoretical insights, I suggest to indicate this in the abstract. Otherwise, the statement remains very vague.
* Section 3.6: In my opinion, the Sec. 3.6 (Theoretical insight) is not an appropriate description of theoretical results in its present form.  The entire technical development is pushed to the appendix, and from the description in the main paper, the problem, setting, and approach do not become clear at all. Further the presented results seems to be rather vague insights (e.g., "we argue..."), rather than theoretically proven/analyzed facts.  In my opinion, if the paper claims to make a theoretical contirbution, this must be properly presented and developed in the main paper.  Otherwise, this part can be removed.  As is, I don't see a clear contribution here (or, at least, I don't understand it from the presentation as is).

(W5) Motivation and choice of the considered baselines, empirical results.

* It is commendable that the authors compare to an alternative approach from literature (Bakker 2020).  However, according to their discussion of related work, there are several approaches for adaptive sampling, including those that are non-RL based.  Based on the presentation, it was not clear to me why the approach by Bakker et al was chosen as the only one.
* Which algorithm is generally considered to be the "best", which one could compare against?
* Would it make sense to also compare to alternative approaches for adaptive sampling (non-RL ones)?
* The results are partially conclusive in the sense that they show superior performance of the proposed method in most, but not all cases.  In particular, for the Gaussian case (which seems to be a standard in compressed sensing), only in one case the proposed adaptive strategy leads to better results (otherwise the "vanilla" random strategy works better!).  It would be nice to see a few more cases, where the proposed method works well.  For example, have the authors considered experiments with trajectory lengths between 20 and 50?
* Also on the second (high-res) data set, the standard random approach with Gaussian measurements outperforms the proposed method.  A hypothesis is formulated as to why the proposed methods performs sub-optimally.  Why is this hypothesis not investigated further?

(W6) The introduction as motivation of this work and to the area of compressed sensing can be improved.  For example:

* Introduction, "other notions of structure have been considered too" -> It would be helpful if the authors could hint at such examples.  Otherwise, the statement is unclear for readers not familiar with Tang et al.  Further, what type of structure is relevant / considered in this paper?
* Introduction: "experimental gains shown by the adaptive methods" is used as a motivation for this study.  However, what experimental gains (what problem, how large is the gain, etc.) is not discussed.  Discussing these (e.g., references and examples from prior work) seems important to better support the relevance of this work.
* The use of "action" in the context of measurements in the introduction is unclear and misleading.  Usually, in the context of reinforcement learning and control, one distinguishes actions (these are input to the controlled system / environment) from measurements.  Here, the introduction states that "measurements are chosen (...) from the space of actions". There are other wordings in the introduction that also mix the terms.  Later on in the paper, when the RL problem is introduced, it becomes clear what is meant.  But currently, and in particular as this paper may also be read by people from the RL community, the introduction is very confusing.  The terms measurements, action, etc. should be better introduced and explained in the context of this work.
* Introduction, "our model can potentially be agnostic to the exact measurement model" -- This seems speculative.  Does the paper support this claim (then this should be said, and how), or is this speculation (when it should probably be avoided, or at least rephrased)?

(W7) Argument in the conclusion

In the conclusion, it is argued that the (sub-optimal) results of the proposed algorithm are to be expected as random measurements are known to be optimal for large problems.  However, the paper proposes a learning approach using random policies.  So, it seems that these policies could, in principle, represent such random measurements.  Hence, I would expect that - with sufficient learning iterations - the RL approach should also be able to discover the random solution, no?  So, in this sense, the RL approach should get close to the known optimum.  This might require also allowing the random policy during evaluation (while, from my understanding, the mean policy is currently being used).



## Minor comments and suggestions

* Related work: Some of the more technical aspects described in Sec. 2 are difficult to follow at the place (Section 2), where the related work is currently presented.  Some of the technical concepts that are used in the discussion, have not been introduced yet..  The authors might want to consider moving the discussion of related work to later in the paper, once the technical problem has been introduced.
* Eq. (1): connection between A and a_t is not made precise.
* The RL problem is addressed with a standard policy gradient approach (VPG).  It would be insightful if the authors could discussed why they opted for this algorithms and whether they have tried more advanced PG algorithms, such as PPO.
* The value network seems rather small with just one hidden layer.  Did the authors experiment with this?

---

> ### Author Response · Authors · 2024-01-24
> **W1**
>
> We thank the reviewer for the thorough and detailed review. We will address the weaknesses highlighted by
> the reviewer in the following.
> >The authors claim that they present a "more general approach" as compared to the work on MRI imaging. However, it is not spelled out in what way their approach is more general.
>
> In the related work section, we mention that by "more general", we mean capable of dealing with both continuous and discrete sensing matrices and observations. More importantly, we tackle both reconstruction and acquisition problems simultaneously thereby contrasting the two-stage training procedure, in which a model is first trained for reconstruction and subsequently for acquisition. With this, we mean that our proposed method is not specifically designed for MRI reconstruction problems and can be applied to different domains. Methods such as the ones proposed in [1, 18] are solutions specific to biomedical imaging problems, and is not immediately obvious from the papers whether the methods can be used in other domains, for example when the action space is continuous. Compared to traditional compressed sensing approaches [8, 4, 15], our method is more general in the sense that it does not require defining a prior over the data. The only prior that we need to define is over the latent space of the VAE, as mentioned in section 3.5 (now 2.5), while the prior over the data is implicitly learned by the decoder.
>
> >"Gelfand widt-based analysis" needs to be explained in Sec. 2.1 if it is relevant. As is, it is not clear what this is and how it relates to the contributions of this work.
>
> Regarding the Gelfand width-based analysis of the paper, we decided to move this theoretical analysis to the supplementary materials, as the main idea of the paper is self-contained without it. We believe that it is of interest to discuss in which sense the adaptive acquisition is useful, since the existing theoretical results, based on Gelfand width analysis, state that "adaptivity is not useful". We clarified this point in the paper and relegate the theoretical discussions to the supplementary materials.
>
> >The simultaneous consideration for acquisition and reconstruction has been done before with supervised learning. The authors distinguish their work especially in that they use RL instead of supervised learning. However, what is the distinctive advantage of RL / their approach is not explained, but should be.
>
> The work from [18] is mentioned in the related work as it is similar to ours, in the sense that they train acquisition and reconstruction networks simultaneously. We also highlight the key differences with our work, namely that they use supervised learning with relatively few acquisition steps, while we use reinforcement learning with more acquisition steps. We do not conjecture on which method is generally better, as that might vary depending on the task.

---

> ### Author Response · Authors · 2024-01-24
> **W2**
>
> >The authors consider the case of noise-free measurements, which is obviously simpler than the noisy case, which is considered in related works on compressed sensing (see Sec. 2.1). If the noisy case is essentially standard, then the consideration of the noise-free problem only appears to be a significant limitation of the relevance of this work. Is this true? Why is the noisy case not considered?
> Similarly, only linear maps are being considered. Is this limiting or still relevant in compressed sensing?
>
> While it is true that the noise-free measurements are simpler than the noisy case, in our experiments we are consistent in the sense that we use the noise-free measurements for all the models. While the noisy measurements are a relevant benchmark for practical applications, we would expect to see a similar trend in performance from all the models. Also, the MRI settings considered in the literature (see [1] as an example) are equivalent to a noise-free formulation, i.e. once a column in the k-space is selected, the correct frequency content of that column is obtained without added noise. Regarding the linear maps, those are common benchmarks in the compressed sensing literature [10, 4, 15].

---

> ### Author Response · Authors · 2024-01-24
> **W3**
>
> >The problem formulation (Sec. 3.1) is not specific to this paper. From my understanding (as non-expert in compressed sensing), the described problem is the typical problem considered in compressed sensing. So, it does not become clear what is the particular question that this paper addresses, and what distinguishes it from prior work. This would be important to understand in order to assess the relevance. Please provide a succinct problem formulation, and then your proposed solution.
>
> We agree with the reviewer that the section title would entail a problem formulation specific to this paper. We intended to instead formulate the generic goal of compressed sensing, which is to get the best possible reconstruction with the least amount of measurements, and then specify how we intend to do so with our framework in section 3.2 (now 2.2), where we explain how we propose to use Reinforcement Learning to learn an optimal measurement acquisition strategy. We changed the title of section 3.1 (now 2.1) to "Compressed Sensing" to make it coherent with the content of the section.
>
> >Goal 1: "possible if we assume a prior about the signal structure, ..." -- What is the prior structure actually assumed in this work?
>
> The concern about Goal 1 is addressed in the first bullet point answer for W1.
> >Eq. (2) seems incorrect, or is unclear. First, \delta is not introduced. Presumably, the authors mean the Dirac distribution. \delta(x_t+1 - x_t) then essentially means that x_t+1 = x_t. This is not the same as x_t being stationary and/or the transition distribution being fixed. In fact, I did not understand from the description what is meant. If the distribution is supposed to be stationary, I believe that the description is incorrect. If the authors mean that x_t+1 = x_t, then this means that the signal is considered constant; that is, there is no (non-trivial) transition dynamics and thus it is unclear why the transition dynamics are introduced and what's the benefit of treating the problem with RL rather than static optimization. In any case, the problem statement is imprecise and unclear here.
>
> With $\delta$ we mean the Dirac distribution, we added a clarification in the revision. What we describe is a way to formulate the adaptive acquisition problem in compressed sensing as a POMDP. In the compressed sensing literature, the case where the underlying signal does not change over time is the most common. Note, however, that we do not have access to such a signal, but only to observations obtained by interacting with the environment. The use of reinforcement learning comes into play as a strategy to select measurements, as opposed to random measurements. While in the specific cases considered in the paper, the underlying state $x$ does not change, our framework can deal with cases in which $x$ evolves over time, so we believe it is appropriate to define $\mathcal{F}$ as a transition distribution.
>
> >Observation model (Sec. 3.3): Because F(.) has previously been introduced as A * x, F would simply be a linear observation model. I find it very misleading that the observation model is first introduced as (i) a distribution (to allow for noise/uncertainty), and (ii) nonlinear map, to then say that only linear and noiseless observations are considered. This is misleading IMO. If the linear map is the one considered, it should be stated like this right away.
>
> We believe that it is important to introduce the framework in a general way, as $F$ is not limited to being linear and deterministic.
>
> >I found the reward unclear as well. First, SSIM should be explained. Then, how can the reward be evaluated as it depends on the true signal x, which is not known in practice, no?
>
> SSIM is a well-known metric available in many ML frameworks such as Pytorch. Therefore, we decided not to describe it in our work. However, we cited the paper that introduced such a metric [17]. Regarding the reward concern, in reinforcement learning the goal is to learn an optimal policy, which uses the reward as a training signal. At deployment, the actions are selected following the learnt policy, and the reward is not needed (unless for evaluation and diagnosis, which is usually done on a test set where the reward signal is available).

---

> ### Author Response · Authors · 2024-01-24
> **W4**
>
> > - Abstract: from the formulation in the abstract, it is unclear what type of results are obtained from the mentioned "theoretical analysis". If there are clear theoretical insights, I suggest to indicate this in the abstract. Otherwise, the statement remains very vague.
> > - Section 3.6: In my opinion, the Sec. 3.6 (Theoretical insight) is not an appropriate description of theoretical results in its present form. The entire technical development is pushed to the appendix, and from the description in the main paper, the problem, setting, and approach do not become clear at all. Further the presented results seems to be rather vague insights (e.g., "we argue..."), rather than theoretically proven/analyzed facts. In my opinion, if the paper claims to make a theoretical contirbution, this must be properly presented and developed in the main paper. Otherwise, this part can be removed. As is, I don't see a clear contribution here (or, at least, I don't understand it from the presentation as is).
>
> Regarding the relevance of the theoretical results, we found that the literature contains seemingly opposing claims about the benefits of adaptivity. On the one hand, among the seminal works in compressed sensing, we can find the result that the adaptive measurements do not improve uniform sample complexity for compressed sensing (see [11, 12]). On the other hand, there are other papers stating the benefits of adaptive measurements in noisy regimes (see for example [7, 9]). Such results raise questions about when and how adaptive acquisition helps. We tried to provide a unified account of this question by looking at the limitations of the current theoretical works and sketching the regimes and types of adaptiveness that will bring about performance improvements. We believe this paints a clearer picture of adaptive acquisition for inverse problems.

---

> ### Author Response · Authors · 2024-01-24
> **W5**
>
> >- It is commendable that the authors compare to an alternative approach from literature (Bakker 2020). However, according to their discussion of related work, there are several approaches for adaptive sampling, including those that are non-RL based. Based on the presentation, it was not clear to me why the approach by Bakker et al was chosen as the only one.
> >- Which algorithm is generally considered to be the "best", which one could compare against?
> >- Would it make sense to also compare to alternative approaches for adaptive sampling (non-RL ones)?
>
> As we mentioned in the related work section, while there are several works using reinforcement learning as an adaptive acquisition strategy, they are mostly focused on the MRI domain. Our goal was to propose a more generic framework, which could be used beyond accelerated MRI. Therefore, we based our method on what was introduced in [1], as it was the most suitable for our use cases, and combined it with the work from [20]. As for which is the best method, that depends on the problem and specific settings. We agree however that a comparison with non-RL methods would strengthen our claims. Therefore, we report a comparison between our approach and the Iterative Soft-Thresholding Algorithm [8] in Appendix D.3.
>
> >The results are partially conclusive in the sense that they show superior performance of the proposed method in most, but not all cases. In particular, for the Gaussian case (which seems to be a standard in compressed sensing), only in one case the proposed adaptive strategy leads to better results (otherwise the "vanilla" random strategy works better!). It would be nice to see a few more cases, where the proposed method works well. For example, have the authors considered experiments with trajectory lengths between 20 and 50?
>
> While we did not train the models specifically on intermediate lengths, we can see the intermediate performances on the graphs in Figure 10, Appendix D.3, which can be used as an indication of what we can expect. For example, the performance after 20 acquisition steps on the model trained on 50 acquisitions is similar to the performance of the model trained on 20 acquisitions. However, in some cases, the performances can be higher or lower (see answer to 3rd bullet point in Additional Feeback).
>
> > Also on the second (high-res) data set, the standard random approach with Gaussian measurements outperforms the proposed method. A hypothesis is formulated as to why the proposed methods performs sub-optimally. Why is this hypothesis not investigated further?
>
> Most reinforcement learning benchmarks for continuous control feature a relatively small action space. For example, the humanoid in the DeepMind control suite [19] has one of the biggest action spaces with 56 joints. It is therefore unclear whether the standard reinforcement learning algorithms are a suitable choice for action spaces of the size considered in our experiments. While it is important to investigate this issue further, we left the analysis for follow-up work.

---

> ### Author Response · Authors · 2024-01-24
> **W6**
>
> >Introduction, "other notions of structure have been considered too" -> It would be helpful if the authors could hint at such examples. Otherwise, the statement is unclear for readers not familiar with Tang et al. Further, what type of structure is relevant / considered in this paper?
>
> Compressed sensing algorithms aim at reconstructing an unknown signal given a set of measurements. Specifically, those measurements represent an under-determined system of equations. Therefore, to find a unique solution to the problem at hand, it is necessary to impose a certain structure on the unknown signal. Sparsity is a common structure used in the compressed sensing literature [10, 6]. However, recently they have been published studies in which the authors leverage different structures on the underlying signals, e.g. model-based compressed sensing [3], manifold models [13, 14], and generative priors [5].
>
> >Introduction: "experimental gains shown by the adaptive methods" is used as a motivation for this study. However, what experimental gains (what problem, how large is the gain, etc.) is not discussed. Discussing these (e.g., references and examples from prior work) seems important to better support the relevance of this work.
>
> The works on accelerated MRI, such as [1] and the other works mentioned in the related work section, are good examples of cases in which the adaptive methods show good experimental gains, in terms of achieving good signal recovery with relatively few measurements. We also provided a theoretical discussion on the benefits of adaptive methods in Appendix A.
>
> >The use of "action" in the context of measurements in the introduction is unclear and misleading. Usually, in the context of reinforcement learning and control, one distinguishes actions (these are input to the controlled system / environment) from measurements. Here, the introduction states that "measurements are chosen (...) from the space of actions". There are other wordings in the introduction that also mix the terms. Later on in the paper, when the RL problem is introduced, it becomes clear what is meant. But currently, and in particular as this paper may also be read by people from the RL community, the introduction is very confusing. The terms measurements, action, etc. should be better introduced and explained in the context of this work.
>
> We agree that using the word action so early in the work was unclear, and we rephrased it to make it more suitable for the introduction.
>
> >Introduction, "our model can potentially be agnostic to the exact measurement model" -- This seems speculative. Does the paper support this claim (then this should be said, and how), or is this speculation (when it should probably be avoided, or at least rephrased)?
>
> When we claim that our method is agnostic to the measurement model, we mean that our model receives as input only the observations and measurements over time, making it unnecessary to provide specific information regarding the sensing operation $F$, as opposed to traditional compressed sensing [10, 15].

---

> ### Author Response · Authors · 2024-01-24
> **W7**
>
> >In the conclusion, it is argued that the (sub-optimal) results of the proposed algorithm are to be expected as random measurements are known to be optimal for large problems. However, the paper proposes a learning approach using random policies. So, it seems that these policies could, in principle, represent such random measurements. Hence, I would expect that - with sufficient learning iterations - the RL approach should also be able to discover the random solution, no? So, in this sense, the RL approach should get close to the known optimum. This might require also allowing the random policy during evaluation (while, from my understanding, the mean policy is currently being used).
>
> In the conclusion, we state that "random measurements are known to be theoretically optimal when a sufficiently long measurement horizon is available", which is not the same as "large problems". If the optimal solution is to take random measurements at all the time steps, then it would be impossible to learn it with reinforcement learning, as there is no learnable structure besides the type of randomness. However, As shown in our experiments, at least for a short acquisition horizon some policies can perform better than random measurements, which means that, at least for a limited acquisition horizon, random measurements are suboptimal. Regarding randomness at deployment, it depends on the application. A deterministic policy is easier to evaluate and randomness at deployment might introduce unpredictable behaviours, which might not be acceptable in some applications.

---

> ### Author Response · Authors · 2024-01-24
> **Minor comments and suggestions**
>
> >- Related work: Some of the more technical aspects described in Sec. 2 are difficult to follow at the place (Section 2), where the related work is currently presented. Some of the technical concepts that are used in the discussion, have not been introduced yet.. The authors might want to consider moving the discussion of related work to later in the paper, once the technical problem has been introduced.
> >- Eq. (1): connection between A and a_t is not made precise.
> >- The RL problem is addressed with a standard policy gradient approach (VPG). It would be insightful if the authors could discussed why they opted for this algorithms and whether they have tried more advanced PG algorithms, such as PPO.
> >- The value network seems rather small with just one hidden layer. Did the authors experiment with this?
>
> - We agree with the reviewer and moved the related work section after the methodology.
> - We added the connection between $A$ and $a_t$ in equation 2.
> - We did provide a comparison with PPO, in appendix D.
> - While it is possible that a better design for the value network could improve performance, in our experiments it seemed to be learning the correct values.

---

### Review · Reviewer_jgrY · 2024-01-04

**Summary Of Contributions:**

This paper introduces an adaptive sensing procedure, aimed at tackling compressed sensing problems.  The approach consists of a combination of supervised learning (for signal reconstruction) and reinforcement learning (for adaptive sensing).  By framing the reinforcement learning problem as POMDP, wherein the hidden state is the true signal, techniques from latent-space RL are leveraged.  The acquisition network, trained via RL, and the reconstruction network, trained via supervised learning, are jointly trained end-to-end.  The paper further explores the use of both auto-encoders and variational auto-encoders as part of the POMDP-RL pipeline.  A theoretical argument for why adaptive sensing may be useful, based on the Gelfand width analysis of Cohen et al, is also provided.  Finally, numerical experiments are conducted on MNIST and Low Dose CT Image and Projection Data.

**Audience:**

Yes

**Claims And Evidence:**

Yes

**Requested Changes:**

Please fix the inconsistent/incorrect use of \citet vs. \citep.

Section 3.6 should be either entirely moved to the appendix, or expanded: in its current state it is too informal to hold any weight as a theoretical insight, and too terse to provide much in the way of intuition.

Without convincing empirical support for the proposed method, which is currently lacking, it’s difficult to support publication.

**Strengths And Weaknesses:**

+ Posing the adaptive measurement acquisition problem as a joint supervised learning and reinforcement learning problem over a POMDP is an interesting and promising approach.

+ The overall architecture presented in Fig. 2 appears sensible and clearly explains the proposed approach.

+ The paper is, for the most part, well written and easy to follow.

+ The paper compares the proposed method against two baselines, AE-R and AE-P.

+ The experimental methodology is well documented in sections 4.1 and 4.2.

- It isn’t clear to me why we should assume access to the generating distribution of the true signals x during the supervised learning phase.  If this generating distribution is not available, how is distribution shift between train and test time accounted for?

- The extension from auto encoders to VAEs seems unnecessary: why not start directly with VAEs?

- The experimental results are not particularly convincing: it does not make sense to me that adaptive design behavior should *degrade* over longer trajectory lengths; furthermore, although random measurements are rate optimal at longer lengths, it’s not clear to me that we shouldn’t expect adaptive measurements to outperform these methods by improving constants.  Finally, there is minimal empirical evidence that VAE-based approaches outperform traditional auto encoders (see table 3 for example).

---

> ### Author Response · Authors · 2024-01-24
>
> We thank the reviewer for the insightful comments and feedback. We fixed the use of citet and citep in the revised version. We address the highlighted weaknesses below.
>
> Regarding Section 3.6, as we stated in other answers, we will move it completely to the supplementary materials and just summarize the key points in a few sentences. To repeat our answer to the other reviewers regarding the relevance of the theoretical results, we found that the literature contains seemingly opposing claims about the benefits of adaptivity. On the one hand, among the seminal works in compressed sensing, we can find the result that the adaptive measurements do not improve uniform sample complexity for compressed sensing (see [12, 11]). On the other hand, there are other papers stating the benefits of adaptive measurements in noisy regimes (see for example [7, 9]). Such results raise questions about when and how adaptive acquisition helps. We tried to provide a unified account of this question by looking at the limitations of the current theoretical works and sketching the regimes and types of adaptiveness that will bring about performance improvements. We believe this paints a clearer picture of adaptive acquisition for inverse problems.
>
> >It isn’t clear to me why we should assume access to the generating distribution of the true signals x during the supervised learning phase. If this generating distribution is not available, how is distribution shift between train and test time accounted for?
>
> The problem of data accessibility and distribution shift is a very important one, but quite generic to the Deep Learning / Generative Models literature and not specific to our work. As pointed out by the reviewer, it is not always the case that the generating distribution of the true signal is available, or more in general, is never the case in the most interesting use cases. What is usually done when creating a dataset, is to collect data samples, and assume that those samples come from the true data distribution and that it will also reflect on the test distribution. However, we stress that this is not a specific problem to our work, and the considered settings are the same as in other compressed sensing works such as [2, 18].
>
> >The extension from auto encoders to VAEs seems unnecessary: why not start directly with VAEs?
>
> As one can see from the experiments, AE and VAE have quite different behaviours depending on the specific settings. We therefore believe that including both results in the manuscript is a valuable contribution.
>
> >The experimental results are not particularly convincing: it does not make sense to me that adaptive design behavior should degrade over longer trajectory lengths; furthermore, although random measurements are rate optimal at longer lengths, it’s not clear to me that we shouldn’t expect adaptive measurements to outperform these methods by improving constants. Finally, there is minimal empirical evidence that VAE-based approaches outperform traditional auto encoders (see table 3 for example).
>
> We agree with the reviewer that the results for long trajectories are not in line with the expectations, and provide some conjectures in sections 4 and 5. However, we would like to stress that the adaptive strategy performs very well for shorter acquisition horizons. Note that such trajectories are "short" only in the context of our paper, and are longer than trajectories considered in other related papers, as pointed out in section 4.3.

---

### Review · Reviewer_aJQC · 2024-01-15

**Summary Of Contributions:**

--

**Audience:**

Yes

**Broader Impact Concerns:**

Not applicable.

**Claims And Evidence:**

Yes

**Requested Changes:**

See response to previous field. Moreover, references missing:

https://openreview.net/pdf/c7e06f5e31c223270b26bc6cf8672959a2f96b1d.pdf

**Strengths And Weaknesses:**

I have the following comments about this work:

-- The paper framing does not sufficiently clarify what are the major theoretical gaps that it aims to fill, and why they are persistent in the literature. In particular, in the second paragraph, it is stated ``It is important to understand the roots of this discrepancy and see if any guidelines can be obtained by revising the existing theoretical results." but it does not state what those gaps are, whether they are specific to some common assumptions underlying pre-existing theoretical results that need to be relaxed, or whether experimental gains that may be observed in practice operate outside the technical conditions typically required. It is not easy to glean the nature of this gap from the discussion of related works. Please underscore this in much more specificity.

--Much greater detail needs to be specified in terms of which prior works have posed this problem class using POMDP framework, and their solution methodologies, to contrast what is new and innovative here about the technical approach.

--The major motivation regarding a theoretical re-examining of prior settings seems a little specious, given that there is relatively little effort given to understanding conditions under which the problem is solvable, what the oracle optimal active learning strategy would be, and how attained performance compares to this. Due to this, one is left with assessing only whether the experiments are sufficient to warrant publication. However, so far as I can tell, strong benchmarks from active learning, such as sequential reduction of uncertainty, and greedy methods, are not compared against.

The approach seems applicable to general inverse problems, but a thorough comparison about how the algorithmic approach contrasts with solution methodologies for Bayesian or frequentist inverse problems is not given. This is missing from the mansucript as well.

---

> ### Author Response · Authors · 2024-01-24
>
> We thank the reviewer for the comments, which we address below. We further studied the proposed missing reference, and we believe that, while sharing marginal similarities with our work, the domain and method of that paper are not related enough. We therefore believe that the reference is not suited for our paper.
>
> >The paper framing does not sufficiently clarify what are the major theoretical gaps that it aims to fill, and why they are persistent in the literature. In particular, in the second paragraph, it is stated ``It is important to understand the roots of this discrepancy and see if any guidelines can be obtained by revising the existing theoretical results." but it does not state what those gaps are, whether they are specific to some common assumptions underlying pre-existing theoretical results that need to be relaxed, or whether experimental gains that may be observed in practice operate outside the technical conditions typically required. It is not easy to glean the nature of this gap from the discussion of related works. Please underscore this in much more specificity.
>
> A detailed analysis is provided in appendix A.
>
> >Much greater detail needs to be specified in terms of which prior works have posed this problem class using POMDP framework, and their solution methodologies, to contrast what is new and innovative here about the technical approach.
>
> To the best of our knowledge, the other works framing the problem with the POMDP framework are already mentioned in the related work, specifically [1, 2, 16].
>
> >The major motivation regarding a theoretical re-examining of prior settings seems a little specious, given that there is relatively little effort given to understanding conditions under which the problem is solvable, what the oracle optimal active learning strategy would be, and how attained performance compares to this. Due to this, one is left with assessing only whether the experiments are sufficient to warrant publication. However, so far as I can tell, strong benchmarks from active learning, such as sequential reduction of uncertainty, and greedy methods, are not compared against.
>
> The reason why we did not add comparisons to the other adaptive methods, such as [1, 18] is because they are specific to the MRI reconstruction problem, while we wanted to introduce a more generic framework. Our work, as opposed to the ones mentioned above, can deal with continuous action space and is not limited to images. However, we added a comparison with a traditional compressed
> sensing approach in Appendix D.3.

---

### Comment · Action_Editor_691V · 2024-01-15
**Additional Feedback**

I received some additional comments and feedback outside of the review system that you may wish to also consider / respond to.

**Overall:**
My general sense reading the paper is that it is currently somewhat sloppy / unpolished in terms of motivations and reasoning. That said, it presents a novel approach with lots of experimental results / ablations on two datasets. I appreciate the theoretical analysis done, although I cannot really evaluate its correctness. The connections to VAEs are interesting, although as it currently stands the derivation of the ELBO (Appendix B) seems incorrect.

Details on ELBO error: In particular, I don't think p(x|z,a,y) in the numerator factorises over timesteps as is done in line 4 to 5. The product symbol should also apply to both numerator and denominator (typo).

**Questions for authors:**
* To my knowledge, the authors are correct in station that they are the first to publish "end-to-end training of reconstruction and acquisition models for active sensing with reinforcement learning". However, some of the reason for this is that Yin et al. (2021) and Bakker et al. (2022) that do end-to-end sensing avoid doing RL because it is unnecessary (and higher variance) in environments with known (differentiable) transitions, which also seems to be the case here.

  * It should also be noted here that Bakker et al. (2022) does adaptive (not active sensing) and Yin et al. (2021) employs a kind of batch-active sensing formulation, rather than doing a single active acquisition at a time. In that sense, the current paper is also different.
  * Note also that the weaker performance on Gaussian w.r.t. random on both MNIST and MAYO may indeed suggest that the RL approach is too high variance.

* For Table 1: do the various stated trajectory lengths refer to separate models trained for these lengths, or are the shorter trajectories cut
from models trained on the full trajectory length?

* Figures 3 and 4: what exactly is on the y-axis here? Is it the cumulative reward seen by the policy during trajectories for the longest-horizon runs?

  * Assuming this interpretation is correct and that models at various trajectory lengths were specifically trained for that length: when comparing Figure 3 to Table 1, it’s interesting that Mean SSIM for the E2E model seems higher at step 50 in Figure 3 than in Table 1. I.e., training on a longer trajectory length seems to on average help the model obtain better performance even at shorter lengths. It’s unclear whether the same is true for T=20, since the y-axis of Figure 3 is not labeled above 0.6.

  * Figure 5: text claims VAE with beta=0.01 is on par with AE, but this seems not true for most of the trajectory in the Gaussian setting (and cannot be seen from the graph in the Radon setting).

* It seems results in tables correspond to a single run only, where standard error is the error computed over various images in the dataset (not across seeds): is that correct?

* Appendix D.4 ablates the policy by having it be: “trained on final reward only”. This performs worse, as expected, presumably because the policy is now doing credit assignment over an entire trajectory using only a single reward. Is there also an ablation where the reconstruction loss only uses the final step? The authors argue that using a multi-step loss is preferable, but I’d like to see this.

**Other notes:**

* Some inline citations should not be inline.

* Table 1 refers to rows signed as M and W, but these are not shown in the table (only in Table 4 of the Appendix).

* Appendix D.2 states: “Bakker et al. (2020) suggests that greedy policies can perform on par or even outperform policies trained with a discounted reward.” Bakker et al.’s actual claim is about greedy policies performing on par or outperforming fully greedy policies with gamma=1. They also show that carefully choosing gamma can ameliorate this effect (e.g., gamma=0.9, as is used in the current paper). That said, the analysis in the current paper surrounding greedy being worse than non-greedy is well supported.

---

> ### Author Response · Authors · 2024-01-24
> **Part 1/2**
>
> We thank the action editor for sharing the additional feedback.
>
> Regarding the typo in the ELBO derivation in line 4 to 5 of Appendix B, we would like to thank the reviewer for spotting it. As the reviewer stated it, it is just a typo, and the rest of the derivations are correct. It can be seen that the sum line 6 is taken over $t$ and includes $q(z_t\mid x, z_{1:t-1}, y_{1:t-1}, a_{1:t-1})$ in the summand. This means that we have assumed that the product symbol should also apply to the denominator. We have corrected the typo in the paper.
>
> Regarding the factorization, the term $p(x\mid a_{1:T}, y_{1:T})$  should be $p(x_{1:T}\mid a_{1:T}, y_{1:T})$, where $x_t$ represents the reconstruction of $x$ at time $t$. This also means that the factorization would work with $p(x\mid z_t, a_t, y_t)$ replaced by $p(x_t\mid z_{t}, a_{t}, y_{t})$. The missing sub-indices are a typo as it was originally included in the last two lines of the derivation in the previous version. We have corrected that in the revised version. We have also included Figure 7 in Appendix B for further clarification.
>
> >To my knowledge, the authors are correct in station that they are the first to publish "end-to-end training of reconstruction and acquisition models for active sensing with reinforcement learning". However, some of the reason for this is that Yin et al. (2021) and Bakker et al. (2022) that do end-to-end sensing avoid doing RL because it is unnecessary (and higher variance) in environments with known (differentiable) transitions, which also seems to be the case here.
> >- It should also be noted here that Bakker et al. (2022) does adaptive (not active sensing) and Yin et al. (2021) employs a kind of batch-active sensing formulation, rather than doing a single active acquisition at a time. In that sense, the current paper is also different.
> >- Note also that the weaker performance on Gaussian w.r.t. random on both MNIST and MAYO may indeed suggest that the RL approach is too high variance.
>
> We agree with the need for further clarification between active and adaptive sensing. In the paper we use them interchangeably, meaning methods that select different observations for different inputs. However, we agree that further clarification and distinction between adaptive and active strategies can improve the readability and clarity of the paper. We added an explanation in section 2.2. Regarding the variance problem, we agree that it’s the most plausible explanation for the difficulty of learning Gaussian acquisition on long trajectories. However, neither of the works using supervised learning report results on long trajectories, which makes it hard to tell whether the high variance problem would be present also in the supervised learning settings.
>
> > For Table 1: do the various stated trajectory lengths refer to separate models trained for these lengths, or are the shorter trajectories cut from models trained on the full trajectory length?
>
> The results in Table 1 refer to separate models, trained for the different lengths. So for each model, there are three versions, trained with trajectory lengths 20, 50 and 100 for Gaussian and 5, 10 and 20 for Radon, respectively.
>
> >Figures 3 and 4: what exactly is on the y-axis here? Is it the cumulative reward seen by the policy during trajectories for the longest-horizon runs?
> >- Assuming this interpretation is correct and that models at various trajectory lengths were specifically trained for that length: when comparing Figure 3 to Table 1, it’s interesting that Mean SSIM for the E2E model seems higher at step 50 in Figure 3 than in Table 1. I.e., training on a longer trajectory length seems to on average help the model obtain better performance even at shorter lengths. It’s unclear whether the same is true for T=20, since the y-axis of Figure 3 is not labeled above 0.6.
> >- Figure 5: text claims VAE with beta=0.01 is on par with AE, but this seems not true for most of the trajectory in the Gaussian setting (and cannot be seen from the graph in the Radon setting).
>
> In figures 3 and 4, on the y axis, we report the SSIM between the ground truth and the reconstructed image after n acquisitions, with n being the value on the x axis. That does not correspond to the reward, which is instead the SSIM at time step n minus the SSIM at time n-1. The observation that at T=50 the model performs better when trained for T=100 than for T=50 is correct, while for T=20 the performance seems to be similar, even though it is not possible to see the exact value from the graph. In Figure 5, we consider the performances at the end of the trajectory, which is on par, while it is true that for intermediate steps AE performs better.

---

> ### Author Response · Authors · 2024-01-24
> **Part 2/2**
>
> >It seems results in tables correspond to a single run only, where standard error is the error computed over various images in the dataset (not across seeds): is that correct?
>
> Yes, this is correct.
>
> >Appendix D.4 ablates the policy by having it be: “trained on final reward only”. This performs worse, as expected, presumably because the policy is now doing credit assignment over an entire trajectory using only a single reward. Is there also an ablation where the reconstruction loss only uses the final step? The authors argue that using a multi-step loss is preferable, but I’d like to see this.
>
> We did not report results for models trained using reconstruction loss only at the last step as those seemed to perform worse in early experiments. Adding these additional experiments is doable for the camera-ready, and we will include them, although given the required simulation time, we could not include them in the current rebuttal version.
>
> >- Some inline citations should not be inline.
> >- Table 1 refers to rows signed as M and W, but these are not shown in the table (only in Table 4 of the Appendix)
>
> We adjusted the citations. We also corrected Table 1, as the worst-case error is only reported in the appendix but we did not correct the table description.

---

### Author Response · Authors · 2024-01-24
**General answer**

We thank the reviewers and the action editors for their detailed and fruitful reviews. We addressed your questions and comments in separate answers. We uploaded a revision, where changes from the previous version are coloured in blue. Below, is a list of papers referred to in the answers.
## 5.1 References
[1] Tim Bakker, Herke van Hoof, and Max Welling. Experimental design for mri by greedy policy search.\
[2] Tim Bakker, Matthew Muckley, Adriana Romero-Soriano, Michal Drozdzal, and Luis Pineda. On learning adaptive acquisition policies for undersampled multi-coil mri reconstruction.\
[3] Richard G Baraniuk, Volkan Cevher, Marco F Duarte, and Chinmay Hegde. Model-based compressive sensing.\
[4] Thomas Blumensath and Mike E Davies. Iterative hard thresholding for compressed sensing.\
[5] Ashish Bora, Ajil Jalal, Eric Price, and Alexandros G Dimakis. Compressed sensing using generative models.\
[6] Emmanuel J Candès, Justin Romberg, and Terence Tao. Robust uncertainty principles: Exact signal reconstruction from highly incomplete frequency information.\
[7] Rui M. Castro. Adaptive sensing performance lower bounds for sparse signal detection and support estimation.\
[8] Ingrid Daubechies, Michel Defrise, and Christine De Mol. An iterative thresholding algorithm for linear inverse problems with a sparsity constraint.\
[9] Mark A. Davenport, Andrew K. Massimino, Deanna Needell, and Tina Woolf. Constrained Adaptive Sensing.\
[10] David L Donoho. Compressed sensing.\
[11] Simon Foucart and Holger Rauhut. A Mathematical Introduction to Compressive Sensing.\
[12] Simon Foucart, Alain Pajor, Holger Rauhut, and Tino Ullrich. The Gelfand widths of lp-balls for 0 ≤ p ≤ 1.\
[13] Chinmay Hegde and Richard G Baraniuk. Signal recovery on incoherent manifolds.\
[14] Chinmay Hegde, Michael Wakin, and Richard Baraniuk. Random projections for manifold learning.\
[15] Hamed Pezeshki, Fabio Valerio Massoli, Arash Behboodi, Taesang Yoo, Arumugam Kannan, Mahmoud Taherzadeh Boroujeni, Qiaoyu Li, Tao Luo, and Joseph B Soriaga. Beyond codebook-based analog beamforming at mmwave: Compressed sensing and machine learning methods.\
[16] Luis Pineda, Sumana Basu, Adriana Romero, Roberto Calandra, and Michal Drozdzal. Active mr k-space sampling with reinforcement learning.\
[17] Zhou Wang, Alan C Bovik, Hamid R Sheikh, and Eero P Simoncelli. Image quality assessment: from error visibility to structural similarity.\
[18] Tianwei Yin, Zihui Wu, He Sun, Adrian V Dalca, Yisong Yue, and Katherine L Bouman. End-to-end sequential sampling and reconstruction for mr imaging.\
[19] Tassa, Yuval, et al. Deepmind control suite.\
[20] Luisa Zintgraf, Kyriacos Shiarlis, Maximilian Igl, Sebastian Schulze, Yarin Gal, Katja Hofmann, and Shimon Whiteson. Varibad: A very good method for bayes-adaptive deep rl via meta-learning.

---

### Author Response · Authors · 2024-02-28

We would like to express our gratitude once more to the reviewers and the Action Editor for their comments and thorough reviews. Given that it has been over a month since we submitted our responses, we kindly inquire if there have been any developments in the review process.

---

### Decision · Action_Editor_691V · 2024-03-25

**Recommendation:** Reject

**Comment:**

None of the reviewers felt the author's revision really addressed the presented concerns.  As one reviewer noted, "The authors have not substantively addressed my concerns from the previous round, but instead attempted to refute their merit."  In the end, there were substantial concerns remaining about whether the paper's claims are convincingly and clearly supported by evidence, which makes the paper unsuitable for TMLR publication in its current form.  See above.

As AE, I do want to apologize to the authors for the substantial delay following their response.  TMLR endeavours for a faster turnaround time.  I am also sad that I was not able to ameliorate the slow response with a more positive outcome.

**Audience:**

Yes.  The problem formulation (even under the noise-free setting) would be of interest to some individuals in TMLR's audience.

**Claims And Evidence:**

The reviewers agreed that support for the claims made in the paper were not substantial, convincing, or clear.  While the authors responded to the reviewers' comments in this regard, there was substantial disagreement on the merits of the concerns, but minimal changes to the manuscript to mitigate the concerns.

Some important areas where claims are not sufficiently supported.
* Problem formulation is not representing the problem addressed in the paper which makes it hard to understand its relationship to other work and to understand its claims (Reviewer fgGo).
* Noisy-case not considered, nor even is the limitation acknowledged (Reviewer fgGo).
* The related work is expanded but it doesn't clearly justify the claims that this work fills an important gap (Reviewer ajQC, fgGO)
* The theoretical insights are too vague to backup the paper's claim of theoretical analysis (Reviewer fgGo)
* Empirical evidence seems incomplete: why does performance degrade over longer trajectories? why is the hypothesis presented not investigated further (Reviewers fgGo and jgrY).

**Resubmission Of Major Revision:**

The authors may consider submitting a major revision at a later time.